# Type-specific dendritic integration in mouse retinal ganglion cells

Yanli Ran[1,2,7], Ziwei Huang[1,2,7], Tom Baden [1,3], Timm Schubert[1,2], Harald Baayen[4], Philipp Berens [1,2,5,6,8], Katrin Franke [1,2,5,8] & Thomas Euler [1,2,5,8 ✉]

Neural computation relies on the integration of synaptic inputs across a neuron's dendritic arbour. However, it is far from understood how different cell types tune this process to establish cell-type specific computations. Here, using two-photon imaging of dendritic $Ca^{2+}$ signals, electrical recordings of somatic voltage and biophysical modelling, we demonstrate that four morphologically distinct types of mouse retinal ganglion cells with overlapping excitatory synaptic input (transient Off alpha, transient Off mini, sustained Off, and F-mini Off) exhibit type-specific dendritic integration profiles: in contrast to the other types, dendrites of transient Off alpha cells were spatially independent, with little receptive field overlap. The temporal correlation of dendritic signals varied also extensively, with the highest and lowest correlation in transient Off mini and transient Off alpha cells, respectively. We show that differences between cell types can likely be explained by differences in backpropagation efficiency, arising from the specific combinations of dendritic morphology and ion channel densities.

[1] Institute for Ophthalmic Research, University of Tübingen, Tübingen, Germany. [2] Centre for Integrative Neuroscience, University of Tübingen, Tübingen, Germany. [3] Sussex Neuroscience, School of Life Sciences, University of Sussex, Brighton, UK. [4] Department of Linguistics, University of Tübingen, Tübingen, Germany. [5] Bernstein Centre for Computational Neuroscience, University of Tübingen, Tübingen, Germany. [6] Institute of Bioinformatics and Medical Informatics, University of Tübingen, Tübingen, Germany. [7] These authors contributed equally: Yanli Ran, Ziwei Huang. [8] These authors jointly supervised this work: Philipp Berens, Katrin Franke, Thomas Euler. ✉email: thomas.euler@cin.uni-tuebingen.de

Across the nervous system, the output signal of a neuron is determined by how it integrates the often thousands of synaptic inputs it receives across its dendritic arbour[1–4]. However, still little is known about how dendritic integration is shaped by differences between neuron types, such as specific dendritic morphology and ion channel complement and density. To investigate type-specific dendritic integration and the key factors driving it, we here use the vertebrate retina, a model system with a clear input–output relationship that can be recorded in a dish[5]. The retina decomposes the visual signal into ~40 feature-specific parallel channels (reviewed in ref. [6]), relayed to the brain by a matching number of retinal ganglion cell (RGC) types[7,8]. RGCs receive their main excitatory drive from the bipolar cells (BCs), which pick up the photoreceptor signal in the outer retina. In addition, RGCs (and BCs) receive inhibitory input from amacrine cells (ACs) (reviewed in ref. [9]), completing the canonical RGC input circuit. Different RGC types differ in morphology[10–12], synaptic connectivity[10,13], and expression of ion channels[14,15].

To explain the emergence of diverse RGC functions, many previous studies have focused on the selective connectivity with presynaptic neurons in the inner plexiform layer (IPL) (e.g. refs. [10,16,17]). Different RGC types arborize in specific layers of the IPL and, hence, receive synaptic inputs from distinct combinations of BC and AC types[10]. This spatiotemporally heterogeneous input provides the basis of type-specific feature extraction[18]. In addition, RGC dendrites may themselves perform complex computations and therefore contribute to the generation of specific output channels, e.g., through their dendritic geometry, and the complement, distribution, and density of passive and active ion channels (reviewed in refs. [1,19]). So far, dendritic processing in the retina has been studied mainly in interneurons (e.g. refs. [20–22]). Despite some theoretical work in this direction (reviewed in ref. [23]), experimental evidence for type-specific dendritic computation and their biophysical mechanisms in RGCs remains limited and is restricted to a few specific types (i.e. direction-selective RGCs[24,25]; On alpha RGCs[26]).

Here, we exploit the unique structure of the IPL to isolate the contributions of type-specific synaptic input profiles from intrinsic cellular mechanisms to elucidate whether RGC types sampling from a similar input space use specific dendritic integration profiles to generate functionally diverse outputs. To this end, we studied the dendritic integration properties of four Off RGC types in the mouse retina that receive excitatory input from a highly overlapping set of presynaptic neurons. To record light stimulus-evoked signals across the dendritic arbour of individual RGCs, we used two-photon $Ca^{2+}$ imaging. We found that these morphologically diverse RGC types differed strongly in their spatio-temporal dendritic integration properties. A biophysical model suggests that the differential dendritic integration in these RGC types arises from the type-specific combination of dendritic morphology and ion channel complement.

## Results

### Estimating local dendritic receptive fields in single RGCs.
To study dendritic integration in different RGC types, we recorded $Ca^{2+}$ signals in response to visual stimulation across the dendritic arbour of individual cells in the ex-vivo, whole-mounted mouse retina using two-photon imaging. For that, we injected individual RGCs with the fluorescent $Ca^{2+}$ indicator dye Oregon Green BAPTA-1 (OGB-1) using sharp electrodes (Methods), resulting in completely labelled individual cells (Fig. 1a). After recording dendritic activity, the cells were 3D-reconstructed (Fig. 1b), allowing us to extract morphological parameters such as dendritic arbour area, branching order and asymmetry. To determine the

cell's dendritic stratification profile across the IPL relative to the ChAT bands, blood vessels labelled with Sulforhodamine 101 (SR101) were used as landmarks (Fig. 1a, b; Methods).

To map dendritic receptive fields (RFs) of RGCs (Fig. 1c, d), we used a binary dense noise stimulus ($20 \times 15$ pixels, 30 µm per pixel) that was centred on the respective recording field. For each recording field ($32 \times 16$ pixels @31.25 Hz), we extracted regions-of-interest (ROIs) along the dendrites using local image correlations (Supplementary Fig. 1a; Methods). Next, we registered the position and distance of each dendritic segment relative to the soma and extracted each ROI's $Ca^{2+}$ signal. To mitigate the effect of low signal-to-noise ratio in some dendritic recordings (Methods), we routinely applied automatic smoothness determination using a Linear-Gaussian Encoding framework[27] to obtain reliable estimates of each ROI's RF (Supplementary Fig. 1b, c).

OGB-1-mediated $Ca^{2+}$ signals have been shown to allow detecting single action potentials and bursts[28–30], as well as subthreshold events[28], suggesting that the resulting $Ca^{2+}$ signal is a useful proxy for membrane voltage. However, other factors, like $Ca^{2+}$ from intracellular stores, $Ca^{2+}$ permeable glutamate receptors, or internal $Ca^{2+}$ buffering may have contributed to the recorded signal. To assess whether $Ca^{2+}$ signal-derived RFs reflect membrane potential-derived RFs, we performed patch-clamp recordings to measure voltage and $Ca^{2+}$ simultaneously while presenting the dense noise stimulus (Supplementary Fig. 2a). We found that the RF estimated from $Ca^{2+}$ responses (in soma or proximal dendrite) were almost identical to those estimated from somatic voltage responses or the spike train (Supplementary Fig. 2b–d). In addition, we found that the gradient (the rate of change) of the recorded $Ca^{2+}$ signals was linearly related to spike rate (Supplementary Fig. 3) in both tOff alpha and tOff mini cells. These results suggest that the light-evoked dendritic $Ca^{2+}$ signals we measured largely reflect $Ca^{2+}$ influx through voltage-gated channels and, hence, membrane depolarisation, consistent with previous findings[31,24].

Finally, we overlaid the RF contours determined from the dendritic $Ca^{2+}$ responses with the cell's morphology (Fig. 1e–g). For each cell, we recorded different dendritic regions at various distances from the soma yielding between 40 and 232 ROIs per cell (Fig. 1h, i; cf. Supplementary Fig. 1c). This enabled us to systematically probe dendritic integration across an RGC's dendritic arbour and link the properties of local dendritic RFs to overall cell morphology.

### Recorded RGCs are clustered into four morphological types.
To compare dendritic integration profiles across RGC types with overlapping excitatory inputs, we focussed on Off RGCs that stratify close to the Off ChAT band (Fig. 2a; Supplementary Fig. 4). We recorded $n = 31$ cells and clustered them into four morphological groups, using four morphological criteria: soma size, arbour asymmetry, arbour density difference, and arbour area following Bae et al.[11] (Fig. 2; Methods). One group likely corresponded to transient Off alpha (tOff alpha) RGCs, as indicated by a large soma and dendritic area (for statistics, see Table 1) and their characteristic stratification profile (compare to 4ow RGCs in the EyeWire database of reconstructed cells of the mouse retina, http://museum.eyewire.org). The second group likely represented the Off mini alpha transient type (tOff mini; ref. [7]): Cells assigned to this group exhibited an IPL stratification profile very similar to tOff alpha cells, but had smaller somata and dendritic areas. The third group resembled the morphology of F-mini[Off] cells[32], exhibiting an IPL stratification profile peaking between the Off ChAT band and the outer IPL border and a small, highly asymmetrical dendritic arbour. Finally, the fourth

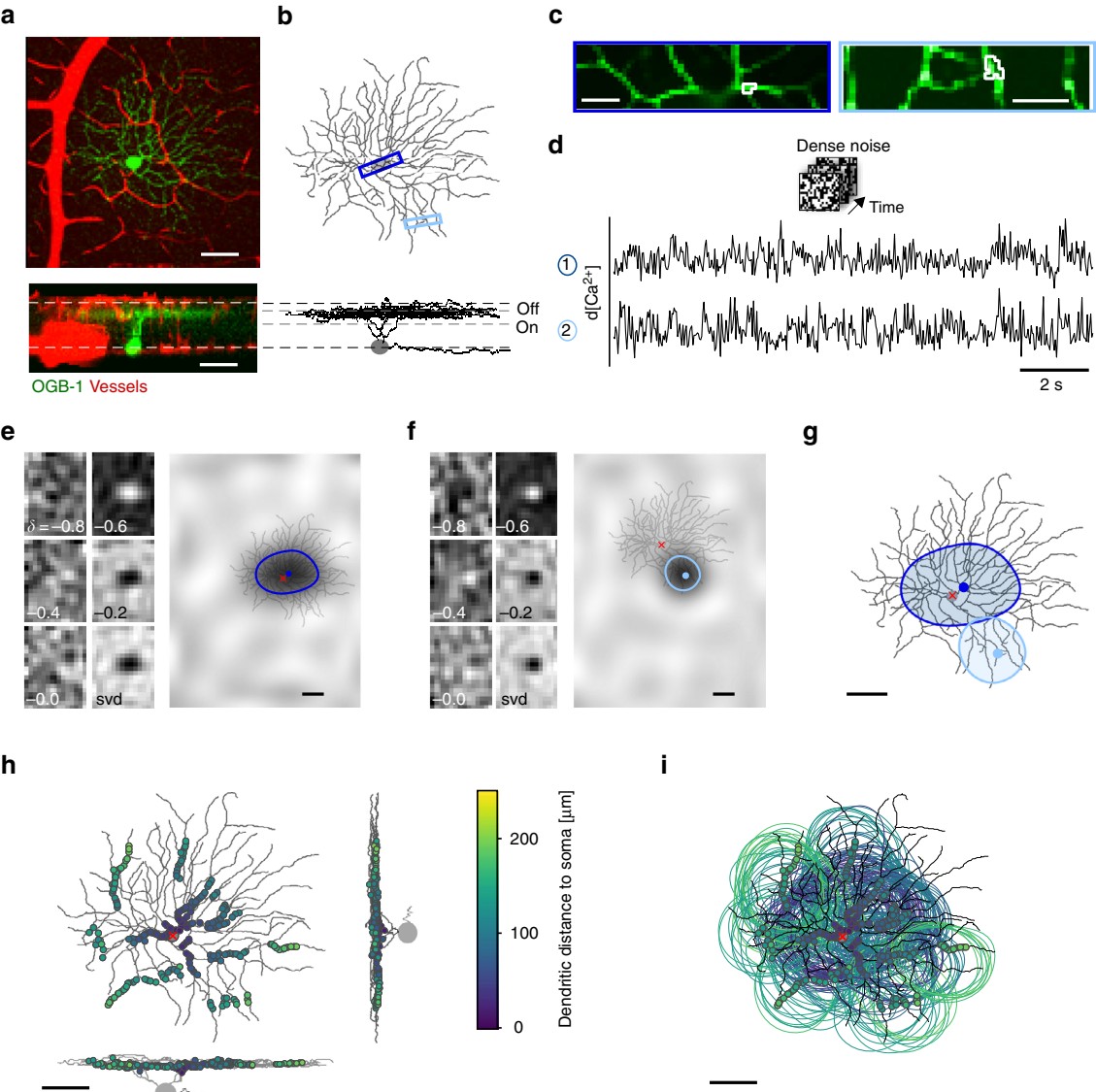

**Fig. 1 Recording dendritic receptive fields (RFs) in individual retinal ganglion cells (RGCs). a** *Z*-projection of an image stack showing an Off-transient RGC filled with the synthetic $Ca^{2+}$ indicator Oregon green BAPTA-1 (OGB-1; green) and blood vessels (red) in top view (top) and as side view (bottom). Dashed white lines mark blood vessels at the borders to ganglion cell layer (GCL) and inner nuclear layer (INL). **b** Reconstructed morphology of cell from **a**. Dashed grey lines between vessel plexi indicate ChAT bands. **c** Example scan fields, as indicated by blue rectangles in **b**, with exemplary region of interest (ROI; white) each. **d** De-trended $Ca^{2+}$ signals from ROIs in **c** during dense noise stimulation (20 × 15 pixels, 30 µm per pixel, 5 Hz). **e** Smooth spatial receptive field (RF) maps from automatic smoothness detection (ASD) for left ROI in **c** at different times ($\delta$, [s]) before an event and singular value decomposition (svd; Methods) map (left). Up-sampled RF map overlaid with the cell's morphology (right; red crosshair indicates soma position), ROI position (blue dot) and RF contour. **f** Like **e** but for right ROI in **c**. **g** RF contours of ROIs from **e**, **f** overlaid on the reconstructed cell morphology. **h** Top- and side-view of example cell with all analysed ROIs ($n$ = 15 scan fields, $n$ = 193 of 232 ROIs passed the quality test; see Methods and Supplementary Fig. 1a, b), shown as dots and colour-coded by dendritic distance from soma. **i** RF contours of ROIs from **h**. Scale bars: **a**, **b**, **e–i** 50 µm, **c** 10 µm.

group displayed a similar IPL stratification profile as sustained Off alpha RGCs (1wt cells in ref. [11]), but had smaller somata and arbour areas. These cells may correspond to the Off sustained ($G_7$) RGCs identified by Baden et al.[7] Here, we refer to them as sustained Off (sOff). In the following, for simplicity, we will refer to these morphological groups as RGC types.

**Dendritic integration profiles vary across RGC types.** Dendrites can process incoming synaptic inputs on a local and a global scale, resulting in rather compartmentalised and synchronised dendrites, respectively[4,33–35]. To investigate whether the four Off RGC types differ with respect to their integration mode, we first assessed how the RF size changed as a function of dendritic

distance to the soma. In tOff alpha cells, local RF area systematically decreased as a function of ROI distance from the soma (Fig. 3a–c; cf. Fig. 1g, i), suggesting that signals in distal dendrites of tOff alpha cells are more isolated and local than those in proximal dendrites. This was not the case in the three remaining RGC types, where RF size remained relatively constant across different positions of the dendritic arbour (Fig. 3a–c; for details, see Supplementary Statistical Analysis). In fact, proximal RFs were significantly larger in tOff alpha cells than in the other RGC types (Fig. 3d), which did not differ systematically in their RF size along their dendrite. Notably, the dendritic RFs of all four Off RGC types were clearly larger than those of Off BCs (Fig. 3c), suggesting that the dendritic RFs we observed largely result from

spatial processing at the level of the RGC dendritic arbour. Together, among the recorded RGC types, dendritic signals in tOff alpha cells are the least spatially synchronised, suggesting that they process dendritic input more locally than the other types.

Synchronisation of dendrites can originate from strong backpropagation of somatic spikes to the dendrites (reviewed in ref. [36]). This is not only expected to increase dendritic RF size but should also shift the RF's centre closer towards the soma or, more precisely, the centre of the dendritic arbour (approximately centre of the cell's total RF). In contrast, for a more isolated dendrite without backpropagation, the RF centre should roughly correspond to the respective ROI position. Therefore, we next analysed for the four RGC types the ROI-to-RF-centre distance (RF offset distance; Fig. 3e, f) as well as the direction of this offset— quantified as the angle between the line from a ROI's centre to the dendritic arbour centre and the line from a ROI's centre to its RF

centre (RF offset angle; Supplementary Fig. 5a). We found that tOff alpha cells displayed small offsets that did not change much across the dendritic arbour, with a substantial fraction of ROIs exhibiting RFs shifted away from the arbour centre (Supplementary Fig. 5b, c). In contrast, the other three RGC types displayed large offsets, with the RF centre strongly shifted towards the centre of the dendritic arbour, which in tOff mini and sOff cells also coincided with the soma (Fig. 3e; Supplementary Fig. 5b, c). Moreover, in tOff mini and sOff cells, offsets increased with dendritic distance from the soma (Fig. 3f). In F-mini$^{Off}$ cells, due to their asymmetrical dendritic arbours, offsets increased with dendritic distance from the arbour centre (Fig. 3f), resulting in an inverted-bell shaped curve. For large dendritic distances, the offsets were significantly different between all pairs of RGC types (Fig. 3g). These results confirm that dendrites of tOff mini, sOff and F-mini$^{Off}$ cells are more synchronised than those of tOff alpha cells, possibly due to backpropagation.

Strongly isolated dendrites, as observed in tOff alpha cells, could allow dendritic computations at a finer spatial scale than the whole cell's RF. Such isolated dendrites are expected to be spatially more independent than the better synchronised dendrites of tOff mini, sOff and F-mini$^{Off}$ cells. To test this prediction, we determined the overlap of RFs for every ROI pair recorded in a single cell (Fig. 4a, b). We then assessed how the overlap changed with dendritic distance and angle between ROIs (Fig. 4b, c; Supplementary Fig. 6). We found localised and spatially independent RFs only in tOff alpha RGCs (Fig. 4a, c; for details, see Supplementary Statistical Analysis). Here, RF overlap decreased substantially with increasing dendritic and angular distance between ROIs, in line with our previous results. In tOff mini cells, RFs showed partial overlap even when the ROIs were located at opposite sides of the dendritic arbour (Fig. 4a, c). For sOff and F-mini$^{Off}$ cells, RFs overlapped substantially, independent of dendritic and angular distance between ROIs. As a result, the RF overlap maps significantly differed between tOff alpha and the other three RGC types, and partially between tOff mini and the remaining two RGC types (Fig. 4d), supporting significant differences in dendritic processing—from more local in tOff alpha to more global in F-mini$^{Off}$.

Together, these results suggest that different RGC types that tap into similar strata of the IPL apply vastly different dendritic integration rules. For example, the dendrites of tOff alpha cells seem to exhibit little backpropagation but reasonably strong forward propagation, integrating RFs from all dendrites symmetrically. This leads to larger proximal than distal RFs and distal RFs with little overlap and displacement. In contrast, the other three RGC types show strong indication for backpropagation across their dendritic arbour, causing distal RFs to be highly overlapping and displaced towards the centre of the dendritic arbour.

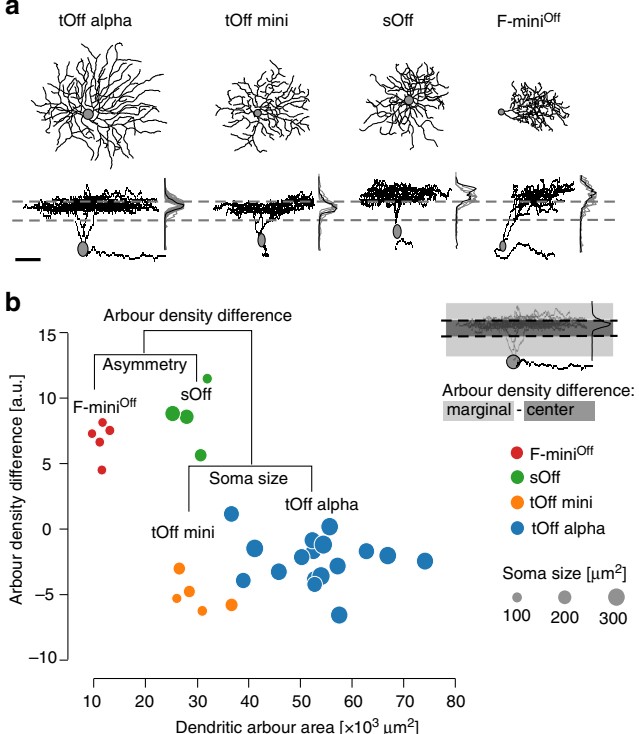

**Fig. 2 Anatomical clustering of recorded RGCs. a,** Top- and side-views of four reconstructed Off RGCs, one of each studied type, with IPL stratification profiles as mean (black) and for all recorded cells of that type (grey). Dashed lines indicate On and Off ChAT bands. **b** Cluster-dendrogram with the morphological parameters used in each clustering step and the resulting RGC groups: $n = 17$ tOff alpha, $n = 5$ tOff mini, $n = 4$ sOff, and $n = 5$ F-mini$^{Off}$. Colours indicate cluster (RGC type), dot diameter represents soma area. Inset: Illustration of arbour density difference measure. Scale bar: **a** 50 μm.

**Temporal dendritic integration varies between RGC types.** Dendritic inputs are not only integrated across space, but also over time. To relate spatial to temporal dendritic integration, we

**Table 1 Morphological parameters describing the dendritic arbours of the clustered RGCs.**

|  | n | Arbour density difference [a.u.] | Area [10³ μm²] | Asymmetry [a.u.] | Soma size [μm²] |
|---|---|---|---|---|---|
| tOff alpha | 17 | −2.41 ± 0.44 | 53.2 ± 2.3 | 44.9 ± 6.3 | 322.5 ± 8.3 |
| tOff mini | 5 | −5.10 ± 0.57 | 29.7 ± 1.9 | 18.9 ± 4.1 | 151.5 ± 14.8 |
| sOff | 4 | 8.58 ± 1.19 | 29.0 ± 1.5 | 14.2 ± 3.7 | 204.0 ± 34.1 |
| F-mini$^{Off}$ | 5 | 6.77 ± 0.62 | 11.5 ± 0.6 | 74.8 ± 6.2 | 102.7 ± 2.2 |

For parameter definitions, see Methods.

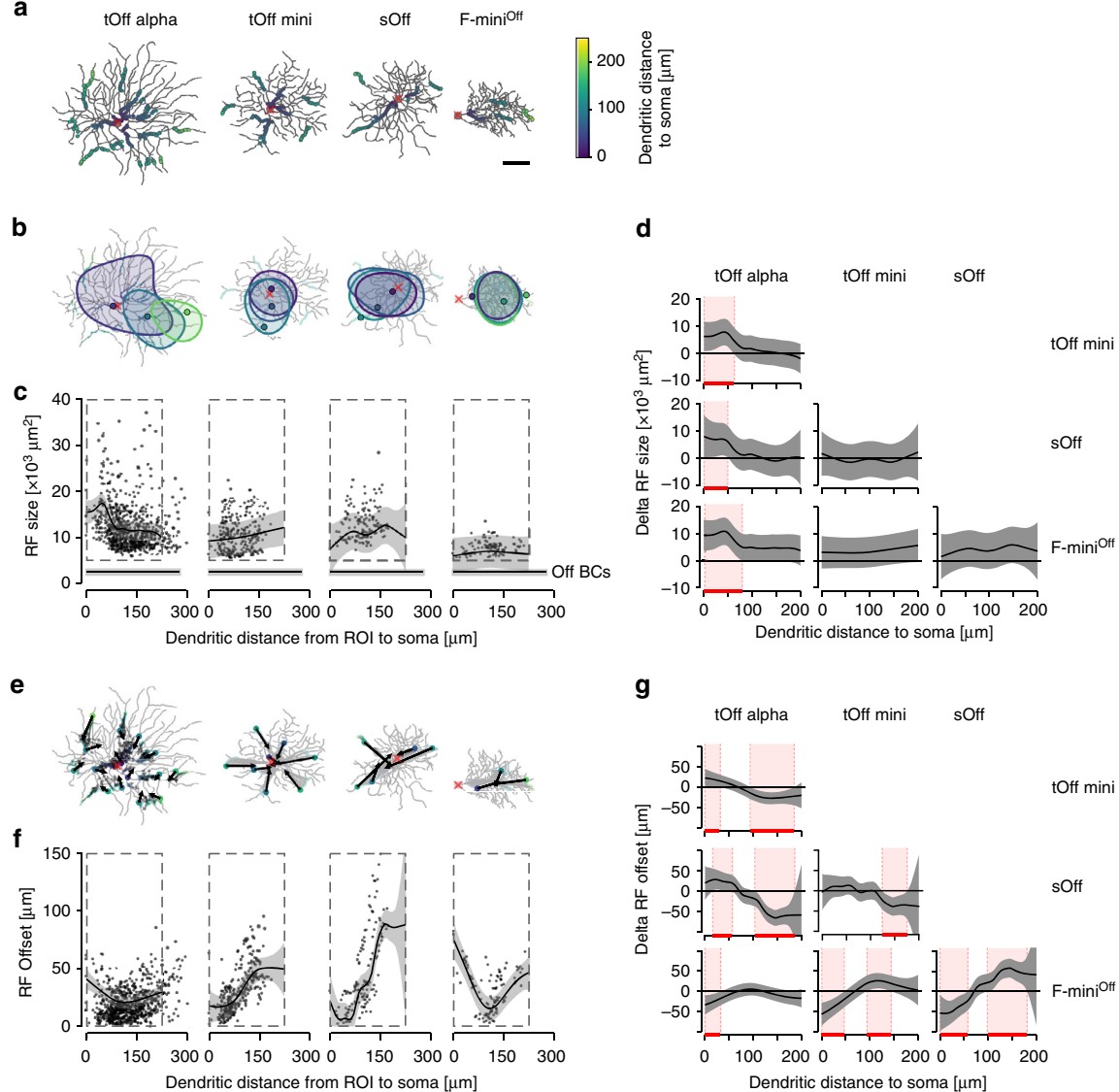

**Fig. 3 Local dendritic RF area and position varies in different RGC types. a** Top-views of the different reconstructed RGC types, overlaid with ROIs that passed the RF quality test. ROI colours indicate dendritic distance to soma. **b** Cells from **a** with three ROIs of increasing distance from soma and corresponding RF contours overlaid (red cross indicates soma position). **c** Dendritic RF area as a function of dendritic distance to soma; data pooled across cells of the same type (see below). Horizontal black line marks mean RF size of Off BCs with s.d. shading in grey ($2360 \pm 1180$ μm², $n = 4242$ ROIs; data from (ref. [37])). **d** Comparison of RF area change with dendritic distance to soma for data marked by the dashed rectangles in **c** between each pair of RGC type. Red shaded areas indicate dendritic sections with significant differences between types (Methods). **e** Cells from **a** with arrows indicating spatial offset between ROI centre and the RF contour's geometrical centre, with arrows pointing at the latter. **f** RF offset of all recorded ROIs as a function of dendritic distance to soma. **g** Like **d** but for RF offset changes for data points inside the dashed rectangles in **f**. Data from tOff alpha ($n = 17\backslash1452\backslash850$ cells\total ROIs\ROIs passing the quality test), tOff mini ($n = 5\backslash387\backslash295$), sOff ($n = 4\backslash208\backslash154$) and F-mini[Off] RGCs ($n = 5\backslash265\backslash126$); for individual cell morphologies, see Supplementary Fig. 4. Scale bar: **a** 50 μm. For details, see Supplementary Statistical Analysis.

next probed the temporal synchronisation of light responses across the dendritic arbour of the four RGC types. For that, we used a chirp stimulus that consisted of a light step followed by frequency and contrast modulations (Methods) and was presented as local (100 μm in diameters) and full-field (800 × 600 μm) version. Notably, F-mini[Off] RGCs did not show any reliable dendritic chirp responses, despite the same ROIs passing our RF quality threshold (Methods). This finding is consistent with earlier observations in this RGC type (cf. x2 cell of Extended Data Fig. 5 in ref. [7]). Therefore, we focussed the following analysis on the remaining three RGC types.

We found that dendritic responses to the local chirp in tOff alpha and tOff mini RGCs were quite similar but differed from those in sOff RGCs (Fig. 5a–c). In the latter, local chirp responses were more sustained than those in the other two types (Supplementary Fig. 7a–c); this difference resonates with sOff cells stratifying slightly more distally (cf. Fig. 2a) and, hence, presumably receiving more input from sustained BC types[16,37]. When presented with the full-field chirp, tOff alpha and tOff mini RGC responses became somewhat more distinct (i.e. to the frequency modulation). This difference was not found in an earlier study[7] but may be related to the fact that in the present study, light stimuli could be precisely centred on the recorded cell. In addition, all three RGC types often showed On-events that were much less frequent for the local chirp (Fig. 5a–c; Supplementary Fig. 7b, d). Similar On-events in Off cells have

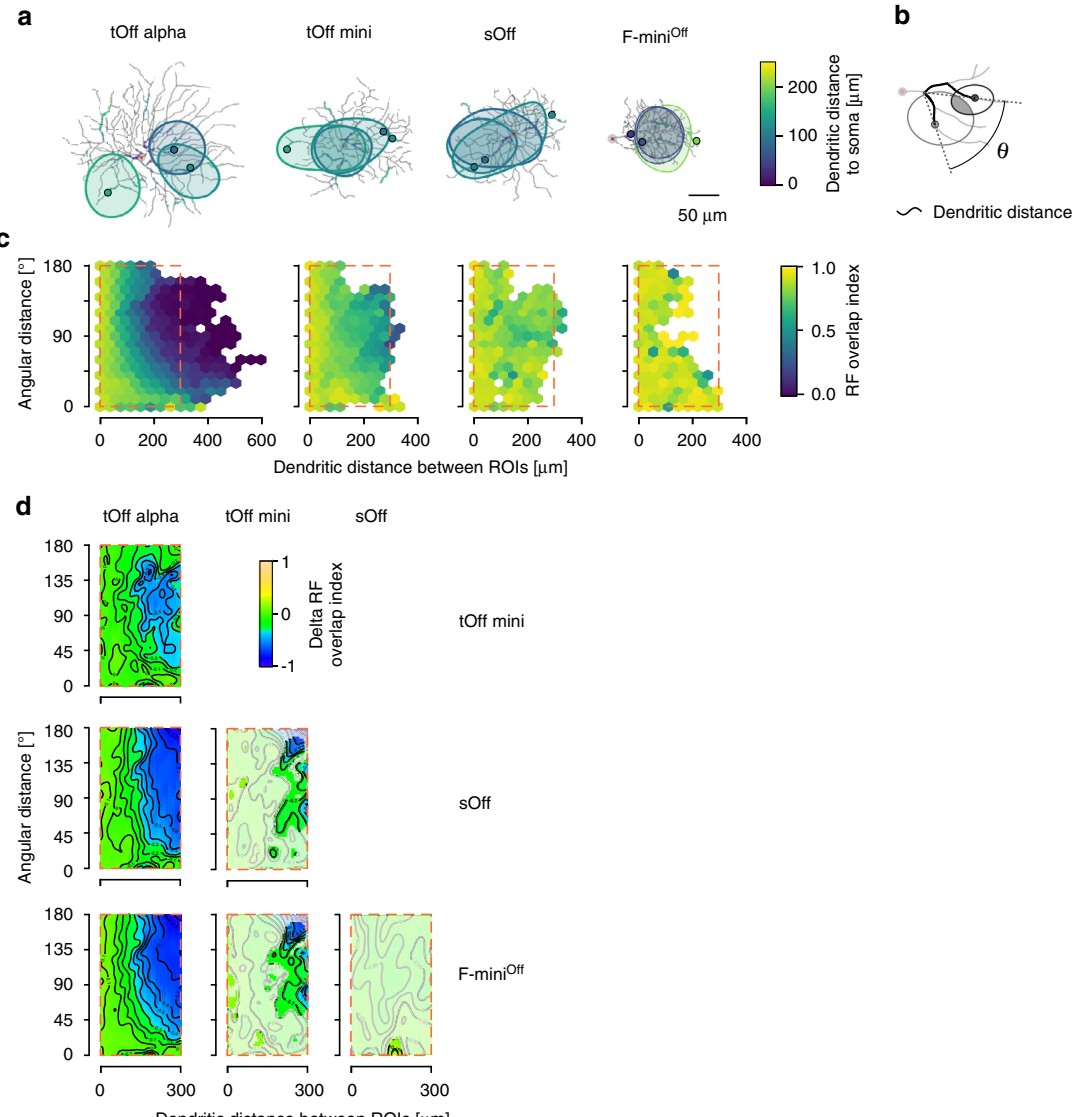

**Fig. 4 Dendritic RF overlap. a** Top-views of the different reconstructed RGC types with RF contours of three ROIs overlaid. **b**, Illustration of dendritic and angular distance ($\Theta$) between two ROIs (measured from the last common branching node) and RF overlap (grey area) of two RF contours (ellipses). **c** Hexagon maps showing the dendritic RF overlap index (colour-coded) as a function of $\Theta$ and dendritic distance for all ROI pairs: tOff alpha ($n = 17\backslash40,777$ cells\ROI pairs), tOff mini ($n = 5\backslash13,524$), sOff ($n = 4\backslash3141$), and F-mini$^{Off}$ ($n = 5\backslash2097$). **d** 2D comparison maps for plot area marked by the dashed red rectangles in **c** for each pair of RGC types. Colour codes difference in RF overlap index, with whitened areas indicating no significant difference. For details, see Supplementary Statistical Analysis.

also been observed in BC responses[37]. In general, differences between full-field and local chirp responses were more pronounced in sOff RGCs (Supplementary Fig. 7e), suggesting that stimulus size had a larger effect on sOff cells compared to the other two types. This could be due to a stronger inhibitory surround or connections to BCs that are more strongly influenced by surround stimulation[37].

To analyse the temporal properties of dendritic integration in these cells, we quantified the correlation of local or full-field chirp responses between ROI pairs across the dendritic arbour (Fig. 5d; Supplementary Fig. 8). In all three RGC types, correlations between ROIs were higher for responses to local than to full-field chirps (Fig. 5d, e), possibly due to surround suppression of the centre excitatory inputs[38]. The decorrelation observed for full-field chirps was especially pronounced in sOff cells (Fig. 5e). In tOff alpha and sOff RGCs, correlation decreased with dendritic and angular distance (Fig. 5d). In

contrast to the other two RGCs, temporal correlation in tOff mini cells was largely independent of dendritic and angular distance (Fig. 5d). In addition, correlation was overall much higher, indicating that dendritic segments in tOff mini cells are temporally more synchronised (cf. Fig. 5c). In tOff alpha and tOff mini cells, lower or higher correlation coincided with smaller or larger RF overlap, respectively. In contrast, sOff RGCs displayed low correlation in their distal dendrites in the presence of highly overlapping RFs across the whole dendritic arbour. This is consistent with the above findings, which suggest that sOff cells may have stronger surround inhibition than the other two RGC types. Because response quality was similar for the RGC types (Supplementary Fig. 9) and differences in temporal correlation between RGC types persisted when applying a more stringent quality criterion (Supplementary Fig. 10; Methods), it is unlikely that they were due to systematic differences in recording quality (i.e. signal-to-noise-ratio).

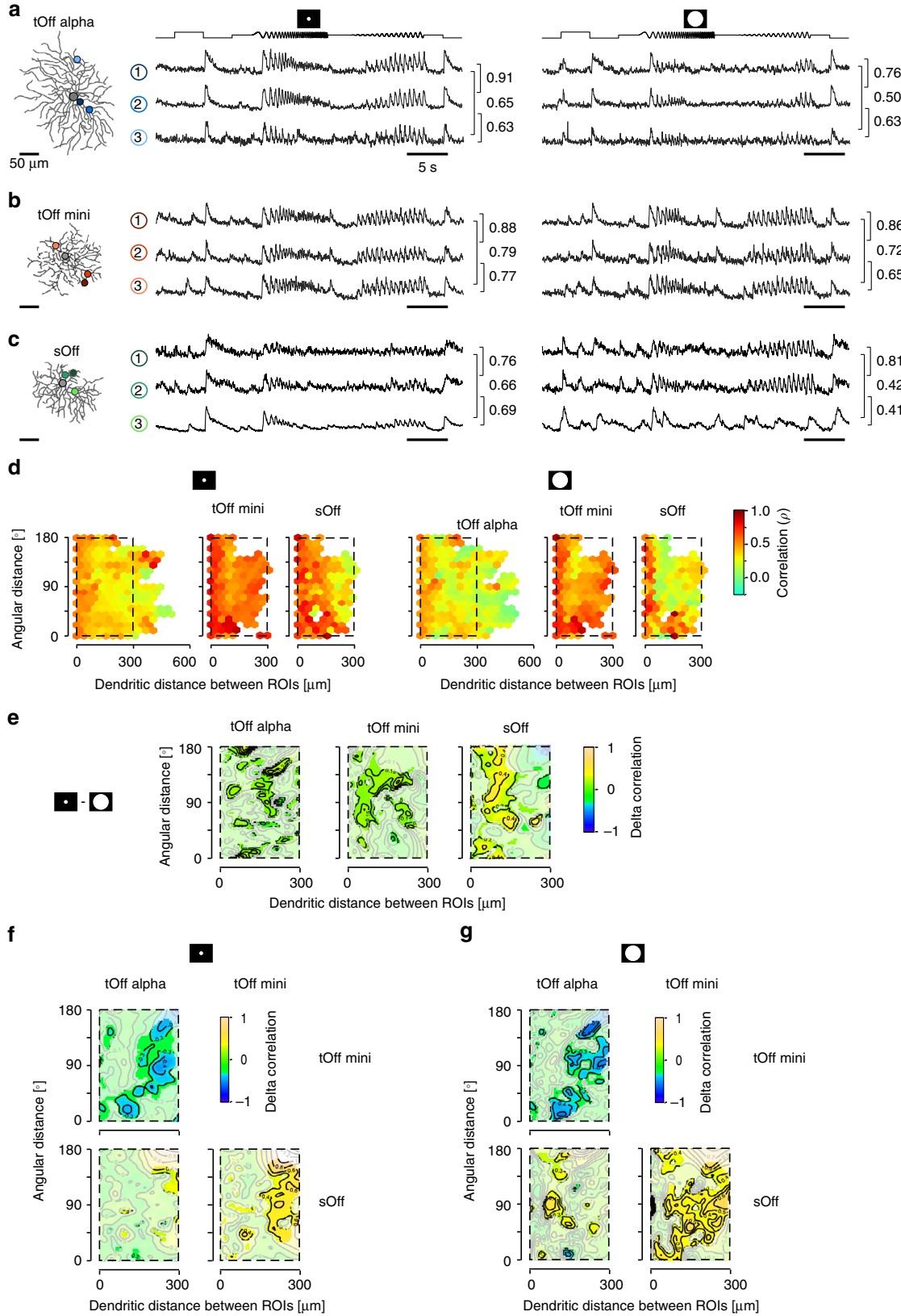

Taken together, our data suggest that spatio-temporal integration is tuned across the RGC dendritic arbour in a highly type-specific manner (Fig. 5f, g). The studied RGC types ranged between two main dendritic integration profiles: The first profile featured strongly isolated dendrites (e.g. in tOff alpha) and may render the cell sensitive to fine visual stimulus structures within the cell's RF. In contrast, the second profile featured strongly synchronised dendrites with highly overlapping RFs (e.g. in tOff mini RGCs) and may tune the cell towards robustly detecting a stimulus independent of its location within the RF.

**Fig. 5 Temporal correlation across dendrites. a** Exemplary response of a tOff alpha RGC to local (middle) and full-field chirp (right) recorded from three ROIs indicated on the reconstructed cell (left). Values next to the traces indicate linear correlation coefficient of the corresponding trace pair. **b, c** Like **a**, but for tOff mini (**b**) and sOff RGC (**c**). **d** Hexagon maps showing response correlations for local (left) and full-field chirp (right) as a function of angular distance and dendritic distance between ROIs for tOff alpha ($n = 17\backslash12,770\backslash13,001$ cells\pairs for full-field\pairs for local), tOff mini ($n = 5\backslash6529\backslash6529$), and sOff RGCs ($n = 4\backslash2622\backslash2557$). Colour encodes correlation. **e** 2D comparison maps for inter-ROI correlation of local and full-field chirp responses for the plot area marked by dashed black rectangle in **d** for each RGC type. Colour codes difference in correlation, with whitened areas indicating no significant difference. **f, g** Like **e**, but for the comparison between cell types for local chirp responses (**f**) and full-field chirp responses (**g**). For details, see Supplementary Statistical Analysis.

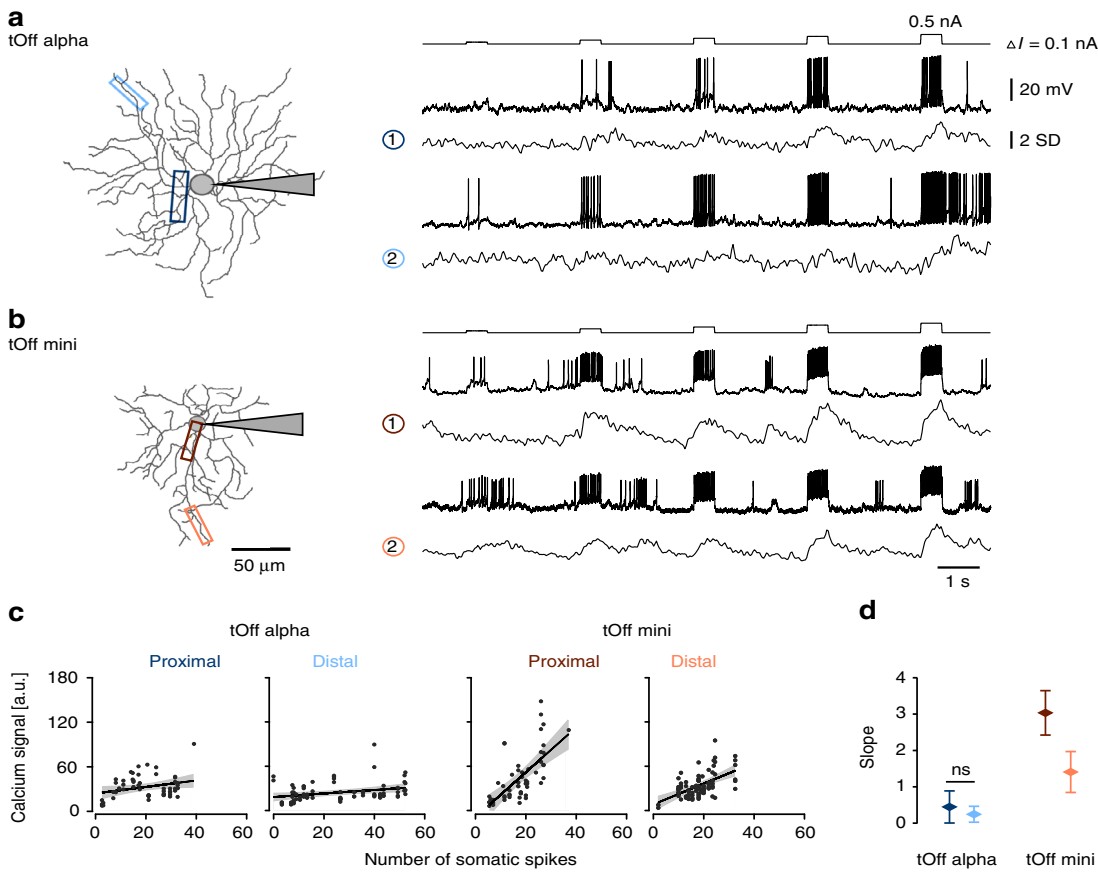

**Fig. 6 Evaluating backpropagation using somatic current injections in tOff alpha and tOff mini cells. a** Left: tOff alpha cell recorded electrically at the soma using whole-cell patch-clamp (shaded triangle represents electrode) while imaging dendritic $Ca^{2+}$ signals. Right: Simultaneously recorded somatic voltage and dendritic $Ca^{2+}$ signals in response to somatic 500-ms current injections (0.1 to 0.5 nA, $\Delta I = 0.1$ nA); $Ca^{2+}$ signals from dendritic regions indicated by the boxes (left). **b** Same as in **a**, but for a tOff mini cell. **c** Dendritic $Ca^{2+}$ signal (as area under the curve) as a function of spike numbers generated in the soma (tOff alpha, $n = 3\backslash55\backslash80$ cells\ROIs for proximal dendrite\ROIs for distal dendrite; tOff mini, $n = 3\backslash55\backslash80$; linear regression and corresponding confidence interval shown as black line and grey shading, respectively). **d** Slope from the linear regression in **c** for proximal and distal dendrites and both RGC types (same colours as in **a–c**). Data are presented as mean with error bars indicating 95% confidence intervals. Except for the slopes estimated based on for proximal and distal dendritic recordings in tOff alpha cells, all the slopes are significantly different across conditions (F(3, 267) = 27.357, $p < 0.0001$. Tukey's method is used for post hoc pairwise comparison. For details, see text and Supplementary Statistical Analysis).

**Differences in backpropagation of spikes between RGC types.** The results so far predict cell type-specific differences in the efficiency of backward propagation of somatic signals, particularly for tOff alpha and tOff mini cells. To test this prediction experimentally, we evaluated the backpropagation of action potentials by whole-cell patch-clamp recordings combined with dendritic $Ca^{2+}$ imaging (Fig. 6). We injected 500-ms current steps (0.1 to 0.5 nA, $\Delta I = 0.1$ nA) into the RGC's soma while simultaneously recording somatic voltage and dendritic $Ca^{2+}$ (Fig. 6a, b). We observed that dendritic $Ca^{2+}$ signals increased much more strongly as a function of the number of evoked action potentials in tOff mini cells than in tOff alpha cells (Fig. 6c). In tOff mini cells, both proximal and distal dendritic $Ca^{2+}$ signals strongly increased with somatic spike count

(slope $= 3.03\backslash1.40$, $r^2 = 0.44\backslash0.31$, $p < 0.001$ for proximal\distal dendrites, $n = 3$ cells), while in tOff alpha cells, we observed much less pronounced increase in $Ca^{2+}$ signals with spike count number (slope $= 0.44\backslash0.24$, $r^2 = 0.08\backslash0.09$, $p = 0.034\backslash0.006$ for proximal/distal dendrites, $n = 3$ cells; Fig. 6d).

Together, these data confirm that backpropagation can be detected with our dendritic $Ca^{2+}$ imaging approach and suggest that backpropagation of action potentials is more efficient in tOff mini compared to tOff alpha cells, supporting the notion that the dendritic arbour of tOff alpha cells is more electrically isolated.

**Simulation reveals mechanisms for type-specific dendritic integration.** The dendritic integration properties of RGC types

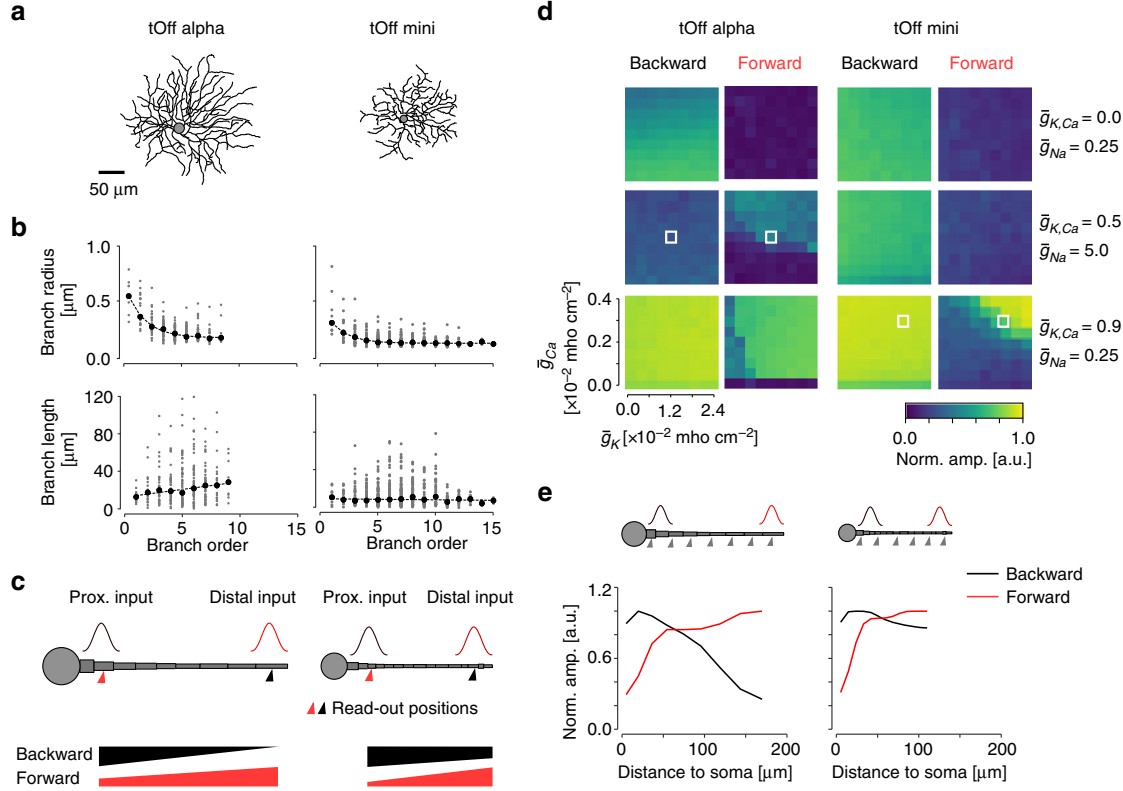

**Fig. 7 Simulation of dendritic signal propagation in tOff alpha and tOff mini RGCs. a** Reconstructed cell morphologies of tOff alpha and tOff mini RGC (same cells as in Figs. 3, 4). **b** Dendrite radius (top) and segment length as functions of branch order (data from http://museum.eyewire.org; $n = 2$ for tOff alpha (4ow); $n = 3$ for tOff mini (4i)). **c** Illustration of the ball-and-stick models used for simulations in **d**, **e**. Simulated inputs at proximal (25 μm to soma) and distal (85% of the total dendrite length to soma) positions indicated as red and black Gaussians, respectively. Respective read-out positions for **d** are indicated below the dendrite. The thickness change of the bars (bottom) corresponds to the decay of forward (red) and backward (black) signal propagation expected from our experimental data. **d** Heat maps showing the signal amplitude at the two read-out positions indicated in **c**, normalised to the amplitude at the respective input position as a function of ion channel density combinations. White boxes indicate channel combinations that are consistent with our experimental results. **e** Normalised signal amplitude at read-out positions along the dendrite as a function of dendritic distance for the channel combinations indicated by boxes in **d**. Generic voltage-gated ($\bar{g}_{Ca}$, $\bar{g}_K$, $\bar{g}_{Na}$) and Ca²⁺ activated ($\bar{g}_{K,Ca}$) conductances were modelled after Fohlmeister and Miller (refs. [67]). For details, see Methods.

may be influenced by morphological features, such as branching pattern, dendritic thickness and segment length, and the complement and distribution of ion channels[35,39]. To understand which of these properties may explain the dendritic integration profiles we observed, we built a simple, morphology-inspired biophysical model and focussed on the effects of the type-specific morphology and dendritic channel densities in tOff alpha and tOff mini cells (Fig. 7a).

To capture the morphological differences between the cell types, we first extracted morphological parameters from a published EM dataset[11]. We found that for tOff alpha cells, dendritic radius decreased systematically with increasing branch order; this decrease was less pronounced in tOff mini cells (Fig. 7b). In addition, dendritic segment length increased with branch order for tOff alpha cells, while it remained constant for tOff mini cells. Based on these differences, we built a ball-and-stick model for each cell type (Fig. 7c). For our simulations, we provided either a proximal or distal input, with read-out positions at the dendritic tip and close to the soma (Methods). Based on the dendritic integration profiles of tOff alpha and tOff mini cells (cf. Figs. 3–6), we hypothesised that (i) in tOff alpha cells, forward propagation (from distal to proximal dendrites) should be stronger than backward propagation and (ii) that backpropagation should be strong in tOff mini cells (Fig. 7c).

To investigate the role of ion channel distribution on dendritic signal propagation, we systematically varied the dendritic density of Ca²⁺-activated K⁺ channels ($\bar{g}_{K,Ca}$) and voltage-gated K⁺, Na⁺, and Ca²⁺ channels ($\bar{g}_K$, $\bar{g}_{Na}$, and $\bar{g}_{Ca}$). Notably, the same combination of channel densities had quite different effects when applied to the two RGC morphologies (compare columns in Fig. 7d and Supplementary Fig. 11), highlighting how strongly the interplay between morphology and channel complement affects a cell's dendritic signal propagation. We found that distinct, cell type-specific sets of ion channel densities were compatible with the experimentally derived hypotheses (Fig. 7d): For the tOff alpha cell model, intermediate $\bar{g}_{K,Ca}$ and high $\bar{g}_{Na}$ and $\bar{g}_{Ca}$ channel densities were required to generate stronger forward propagation compared to backward propagation (Fig. 7d, e). For the same channel densities, forward propagation in modelled tOff mini cell was so low that distal inputs were almost completely extinguished before reaching the proximal dendrite. In contrast, with higher $\bar{g}_K$ and lower $\bar{g}_{Na}$ densities, tOff mini cells showed strong backward and substantial forward propagation, in line with our hypothesis (Fig. 7d, e).

Together, these results suggest that morphology alone does not explain the experimentally observed differences between the two cell types. Instead, our model indicates that differences in

dendritic channel densities may be responsible for the distinct dendritic integration profiles in RGCs.

## Discussion

Here, we studied dendritic integration in four types of mouse Off RGC (tOff alpha, tOff mini, sOff, and F-mini[Off]), which have their dendrites in overlapping strata of the IPL and, hence, receive highly overlapping sets of synaptic input. Recordings of local, light-evoked dendritic $Ca^{2+}$ signals and compartmental modelling revealed surprising differences between the cells' spatio-temporal dendritic integration. What could these distinct integration rules be good for in terms of visual computations?

In tOff alpha RGCs[11], as the distance from the soma increased, RF area decreased and dendritic RFs became increasingly non-overlapping, with minimal offset between recording site and respective RF centre. In addition, activity on different dendritic branches was only moderately correlated. The more isolated, independent dendritic segments in tOff alpha cells may help them to detect fine structures of visual stimuli and support visual computations relying on spatial resolution below the RF of the entire cell. This is reminiscent of what has been reported about On alpha cells, which possess nonlinear RFs and respond to patterns that contain local structures finer than the cell's RF centre[26]. In contrast, in tOff mini and sOff RGCs[7], RFs overlapped extensively and changed little in area, while their centres were systematically shifted towards the soma. In addition, the timing of responses was highly correlated across tOff mini dendrites, suggesting they may reliably detect stimuli independent of their location within the RF. For sOff RGCs, the temporal correlation between the activity of different dendritic branches decreased strongly for larger stimuli, suggesting that the cell's computational properties change as a function of stimulus size. A possible mechanism for the dependence of temporal correlation on stimulus size—not only in the sOff cells—may be shunting inhibition provided by lateral AC circuits kicking in as stimulus size increases[38,40]. F-mini[Off] cells[32] were similar to tOff mini and sOff RGCs with some particularities related to the high asymmetry of their dendritic arbour. Our morphologically inspired biophysical model revealed that morphological difference alone cannot explain these experimentally observed dendritic integration profiles; instead, distinct combinations of morphology, ion channel complements, and densities are required.

Dendritic integration rules have been studied extensively in the cortex (e.g. refs. [41–43]). In the retina, mainly interneurons have been at the centre of interest: For example, it has been suggested that horizontal cells[20] and A17 ACs[22] provide locally computed feedback by confining signals within single varicosities. Likewise, starburst AC dendrites compute the direction of motion dendrite-wise by dividing their dendritic arbour into isolated sectors which contain 15–20 varicosities each[44,45]. In RGCs, dendritic integration has been studied in direction-selective (DS) RGCs, where intrinsic properties of their dendritic arbour[25,46], partially their asymmetry[47], as well as the spatial arrangement of their synaptic input (reviewed in ref. [48]) contribute to the generation of DS output. Reminiscent of our findings in tOff alpha cell, the dendritic arbour of DS RGCs is functionally partitioned, with the DS mechanism replicated across the dendritic arbour, such that local motion within the cell's RF can cause a robust spiking response[24,49].

We chose to focus on four types of Off RGCs because they are expected to receive excitatory inputs from overlapping sets of BC types. Nevertheless, due to small differences in dendritic stratification depth, they make connections with partially different sets of BCs: tOff alpha cells contact dominantly transient type 3a and 4 BCs, while sOff cells likely contact dominantly the more sustained type 1 and 2 BCs[10,11,16]. In line with this, we found that

the dendrites of tOff alpha cells exhibited more transient responses than those from sOff cells. Since tOff mini RGCs co-stratify with tOff alpha RGCs, they potentially receive excitatory inputs from the same BC types and thus should exhibit similar response properties. Indeed, tOff alpha and tOff mini cells showed similar responses to local chirps. Nevertheless, they may be differentially modulated by type-specific connectivity to ACs. In line with this, the two cell types showed more distinct responses to full-field chirps.

In principle, the interaction of excitation from BCs and inhibition from ACs may attenuate the excitatory inputs and affect dendritic integration[40], raising the possibility that the observed type-specific differences could at least partially result from type-specific microcircuit connectivity rather than mainly from cell-intrinsic properties as suggested above. For instance, it has been reported that the responses of tOff alpha RGCs are shaped by the properties of electrically coupled ACs[50]. Shunting by such electrical synapses could contribute to the observed portioning of the tOff alpha cell's dendritic arbour. While such synaptic interactions are expected to contribute to some degree, our simulation results, in combination with our experimental data on dendritic propagation efficiency, indicate that the observed differences in RGC dendritic integration profiles may heavily rely on cell-intrinsic mechanisms.

Apart from contributions of the upstream microcircuit, dendritic integration is mainly determined by a combination of morphological features and passive and active membrane properties, which can differ significantly between RGC types (reviewed in ref. [51]). In some RGC types like the tOff alpha, for instance, the dendritic diameter becomes smaller and dendritic segment length gets longer with increasing branch order. This, in turn, results in a higher axial resistance and shorter propagation distance for more distal dendritic signals. In other RGC types like tOff mini, however, dendritic diameter and segment length does not systematically change with increasing branch order. In addition, a variety of ion channels, including $Ca^{2+}$-activated $K^+$ channels, hyperpolarization-activated cyclic nucleotide-gated (HCN) channels, and voltage-gated $K^+$, $Na^+$, and $Ca^{2+}$ channels, have been found in RGC dendrites, differing in density and dendritic locations between cell types[31,51].

An earlier theoretical study suggested that alpha RGCs—with their large dendritic arbours, thick and short proximal but thin and long distal branches[52]—feature independent dendritic regions[35]. In contrast, RGCs with constant dendritic diameter and branch length across their dendritic arbour are thought to produce densely coupled dendritic regions. In these RGCs, their morphology could enable more efficient dendritic back-propagation and therefore lead to the synchronisation of dendritic signals[53]. Indeed, we observed more independent dendritic regions in tOff alpha cells, but more spatially synchronised dendritic regions in tOff mini, sOff and F-mini[Off] cells. In tOff mini and tOff alpha cells, their forward and backward propagation were differentially modulated by the same combinations of ion channel densities, confirming that dendritic morphology is a key determinant of dendritic signal propagation efficiency. However, our simulation results suggest that the dendritic integration properties of tOff alpha and tOff mini RGCs could not be explained by dendritic morphology alone but require dendritic ion channels in agreement with earlier simulation studies[54]. One possible reason might be that for most RGCs, action potentials generated in the soma can back propagate to the dendritic arbour[55], which needs dendritic ion channels to enable the efficient backpropagation[55,56].

Our simulation results are based on highly simplified ball-and-stick models of RGC dendrites, as these allowed us to focus on the principles of dendritic integration. Obviously, these models come with several caveats and possibilities for future extensions: First, the

**Table 2 Software used and repositories for custom scripts and data.**

| Part | Description (link) | Company/Author | Item number (RRID) |
|---|---|---|---|
| ScanM (v2.04) | 2P imaging software running under IGOR Pro | Written by M. Müller (MPI Neurobiology, Martinsried), and T.E. | |
| pClamp (v10.6) | Electrophysiology Data Acquisition & Analysis Software | distributed by Molecular Devices LLC | |
| IGOR Pro (v6) | https://www.wavemetrics.com | Wavemetrics, Lake Oswego, OR | IGOR Pro v6 (SCR_000325) |
| Python (v3.6.7) | http://www.python.org/ | | |
| R (v2.3) | The R project http://www.r-project.org/ | | |
| QDSpy (v0.77) | Visual stimulation software https://github.com/eulerlab/QDSpy | Written by T.E, supported by Tom Boissonnet (EMBL, Monterotondo) | (SCR_016985) |
| Scikit-learn (v0.20.0) | Software package for Python | ref. [68] | |
| Scikit-Image (v0.14.2) | Software package for Python | ref. [69] | |
| Itsadug (v2.3) | Software package for R | ref. [70] | |
| Mgcv (v1.8-24) | Software package for R | ref. [65] | |
| emmeans (1.4.4) | Software package for R | ref. [71] | |
| NEURON (v7.7.0) | Simulation environment for modelling individual neurons and networks of neurons | ref. [66] | |
| Custom scripts and data | http://retinal-functomics.net/ | | |

morphological parameters we used were extracted from EM data[11], where tissue can shrink due to chemical fixation, such that we may have underestimated the axial conductance based on dendrite diameter. Second, it has been reported that the branching pattern is an important variable for determining propagation efficiency of dendritic signals, mainly because of the diameter changes at branch points[57]. Third, the density and complement of ion channels can vary along the dendrite[51,58], raising the possibility that spatially varying ion channel densities would allow for more refined control over dendritic computations. Finally, dendritic signalling is driven by the complex interaction of excitatory and inhibitory inputs (as already mentioned above) and the locations of the respective synapses, which will require more precise connectomic studies of the cell types and microcircuits in question. A more realistic model incorporating these aspects could allow additional insights into the mechanisms underlying the observed spatio-temporal dendritic integration rules.

## Methods

**Animals and tissue preparation.** Mice used in this study were purchased from Jackson Laboratory and housed under a standard 12 h day/night cycle with 22 °C, 55% humidity. For all experiments, mice aged 5–8 weeks of either sex were used. We used the transgenic mouse line B6;129P2-Pvalb^{tm1(cre)Arbr}/J (PV, JAX 008069, The Jackson Laboratory, Bar Habor, ME; refs. [59]) cross-bred with the red florescence Cre-dependent reporter line Gt(ROSA)26Sor^{tm9(CAG-tdTomato)Hze} (Ai9^{tdTomato}, JAX 007905) for all recordings of tOff mini, sOff and F-mini^{Off} cells (n = 25 animals). For some alpha RGC recordings, we also used the wild-type line (C57Bl/6J, JAX 000664, n = 3 animals), as alpha RGCs can be easily targeted due to their large soma size. Also for the electrophysiological recordings, we used wild-type mice (C57Bl/6J, n = 6 animals). All animal procedures were approved by the governmental review board (Regierungspräsidium Tübingen, Baden-Württemberg, Konrad-Adenauer-Str. 20, 72072 Tübingen, Germany) and performed according to the laws governing animal experimentation issued by the German Government.

Mice were dark adapted ≥2 h before tissue preparation, then anaesthetised with isoflurane (Baxter, Hechingen Germany) and killed with cervical dislocation. The eyes were quickly enucleated in carboxygenated (95% $O_2$, 5% $CO_2$) artificial cerebral spinal fluid (ACSF) solution containing (in mM): 125 NaCl, 2.5 KCl, 2 $CaCl_2$, 1 $MgCl_2$, 1.25 $NaH_2PO_4$, 26 $NaHCO_3$, 20 glucose, and 0.5 L-glutamine (pH 7.4). After removing cornea, sclera and vitreous body, the retina was flattened on an Anodisc (0.2 µm pore size, GE Healthcare, Pittsburgh, PA) with the ganglion cell side facing up and then transferred to the recording chamber of the microscope, where it was continuously perfused with carboxygenated ACSF (at 35 °C and 4 ml min⁻¹). All experimental procedures were carried out under very dim red light.

**Loading of individual cells with calcium indicator.** To visualise blood vessels and avoiding them when filling individual RGCs, 5 µl of a 50 mM sulforhodamine-101 (SR101, Invitrogen/Thermo Fisher Scientific, Dreieich, Germany) stock solution was added per litre ACSF solution. Sharp electrodes for single-cell injection were pulled on a P-1000 micropipette puller (Sutter Instruments, Novato, CA) with resistances ranging between 70 and 130 MΩ. Oregon Green BAPTA-1 (OGB-1, hexapotassium salt; Life Technologies, Darmstadt, Germany; 15 mM in water), a synthetic $Ca^{2+}$ indicator dye with high $Ca^{2+}$ affinity ($K_D = 170$ nM; Invitrogen) and comparatively fast kinetics[29], was loaded into individual RGCs using the single-pulse function (500 ms, −10 nA) of a MultiClamp 900A amplifier (Axon Instruments/Molecular Devices, Wokingham, UK). To allow the cells to completely fill and recover, we started recordings 1 h post injection.

**Two-photon imaging and light stimulation.** A MOM-type two-photon microscope (designed by W. Denk, MPI, Martinsried; purchased from Sutter Instruments/Science Products) as described previously[60] was used for this study. Briefly, the system was equipped with a mode-locked Ti:Sapphire laser (MaiTai-HP DeepSee, Newport Spectra-Physics, Darmstadt, Germany), green and red fluorescence detection channels for OGB-1 (HQ 510/84, AHF, Tübingen, Germany) and SR101/tdTomato (HQ 630/60, AHF), and a water immersion objective (W Plan-Apochromat 20×/1,0 DIC M27, Zeiss, Oberkochen, Germany). For all scans, we tuned the laser to 927 nm, and used a custom-made software (ScanM, by M. Müller, MPI, Martinsried, and T.E.) running under IGOR Pro 6.3 for Windows (Wavemetrics, Portland, OR). Time-elapsed dendritic signals were recorded with 64 × 16 pixel image sequences (31.25 Hz). High-resolution morphology stacks were acquired using 512 × 512 pixel image stacks with 0.8 or 1.0 µm z steps.

Light stimuli were projected through the objective lens[60]. We used two alternative digital light processing (DLP) projectors: a K11 (Acer, Ahrensburg, Germany) or a LightCrafter E4500 MKII (Texas Instruments, Dallas, TX; modified by EKB Technologies Ltd., Israel). Both were equipped with light-emitting diodes (LEDs)— green (575 nm) and UV (390 nm)—that match the spectral sensitivities of mouse M- and S-opsins (for details, see refs. [7,61]). Both LEDs were intensity-calibrated to range from $0.1 \times 10^3$ (black background) to $20.0 \times 10^3$ (white full field) photoisomerisations $P * s^{-1}$ cone⁻¹. The light stimulus was centred before every experiment, ensuring that its centre corresponded to the centre of the microscope's scan field. For all experiments, the tissue was kept at a constant mean stimulator intensity level for ≥15 s after the laser scanning started and before light stimuli were presented.

Light stimuli were generated and presented using the Python-based software package QDSpy (Table 2). Three types of light stimuli were used:

(1) Binary dense noise (20 × 15 matrix of 30 µm per pixel; each pixel displayed an independent, balanced random sequence at 5 Hz for 5 min) for spatio-temporal receptive field (RF) mapping. The pixel size was chosen to be slightly smaller than the RF centre of single BCs (38–68 µm in diameter; ref. [37]), allowing to estimate RGC dendritic RFs at single-BC resolution.
(2) Full-field (800 × 600 µm) chirp, consisting of a bright step and two sinusoidal intensity modulations, one with increasing frequency (0.5–8 Hz) and one with increasing contrast.
(3) Local chirp; like (2) but with a diameter of 100 µm.

**Simultaneous recordings of somatic voltage and dendritic $Ca^{2+}$.** For the simultaneous recording of somatic voltage and dendritic $Ca^{2+}$ in response to light stimuli or current injections (0.1–0.5 nA, $\Delta I = 0.1$ nA), we performed whole-cell patch-clamp recordings (electrode resistance, 7–15 MΩ). In addition to 200 µM of

OGB-1, the intracellular solution contained (in mM): 120 K-gluconate, 5 NaCl, 10 KCl, 1 MgCl$_2$, 1 EGTA, 10 HEPES, 2 Mg-ATP, and 0.5 Tris-GTP, adjusted to pH 7.2 using 1 M KOH. Before the whole-cell recordings, liquid junction potentials of 15 mV were corrected with the pipette offset function of the Axopatch 200B amplifier (Molecular Devices LLC). To allow the dendrites to fill with OGB-1, cells were kept in the whole-cell mode for ~5 min before the start of the recordings. In some cases, we performed extracellular (cell-attached) recordings. Here, the cells were first injected with OGB-1 (as described above) and then targeted with ASCF-loaded electrodes. All electrophysiological data were digitised at 10 kHz using the pClamp software (Molecular Devices GmbH) and Bessel-filtered at 2 kHz. All Ca$^{2+}$ signals were imaged at 31.25 Hz (64 × 16 pixel image sequences).

**Reconstruction of cell morphologies.** Directly after the recording, the complete dendritic morphology of the RGC was captured by acquiring a high-resolution stack. In case the cell was not bright enough to see all branches in detail, a second dye-injection was performed. Using semi-automatic neurite tracing[62], we obtained cell skeletons of the recorded RGCs. If necessary, we de-warped image stack and traced cell, as described earlier[7]. All further analysis, such as the extraction of morphological parameters (see below), was done using custom Python scripts.

**Relating recording positions to cell morphology.** As the full dendritic morphology could not be imaged during Ca$^{2+}$ recordings, recorded dendrites (i.e. regions of interest (ROIs), see below) were not necessarily well-aligned with the cell morphology reconstructed later. Based on the relative position of each recording field, a region 9 times larger than this recording field was cropped from the reconstructed morphology and z-projected. Next, the recording field was automatically aligned to this cropped region using `match_templates` from *scikit-image* (Table 2). The centre coordinates of all ROIs in that recording field were then calibrated to the closest dendritic branch based on their Euclidean distance. In rare cases, when automatic matching failed, the matching was done manually.

**Morphological parameters and hierarchical clustering.** To morphologically cluster the RGCs as described by Bae et al.[11], we had to determine the relative position of the two ChAT bands, the dendritic plexi of the starburst ACs[63]. For this, the blood vessel plexi in GCL and INL served as landmarks. With their positions defined as 0 and 1, the relative IPL depth of the On and Off ChAT bands is 0.48 and 0.77, respectively, as shown earlier[7,37]. The following parameters were extracted for each cell:

To determine the marginal-central arbour density difference, we defined the central IPL as the portion between the ChAT bands and the remainder (On ChAT band to GCL, Off ChAT band to INL) as marginal IPL (cf. Fig. 2b, inset). The marginal-central arbour density difference was calculated using the sum of the dendritic length in central IPL minus the sum of the dendritic length located in marginal IPL.

Dendritic arbour area was calculated as the area of the tightest convex hull containing the z-projected dendritic arbour.

Asymmetry of the dendritic arbour was calculated as the distance between the centre of mass of dendritic density and the soma position.

Soma size was defined as soma area. For this, the image frame in which the soma appeared the largest was used.

Dendritic distance between ROIs was defined as the shortest distance along the dendrite between two ROIs.

Angular distance between ROIs was defined as the positive angle between two ROIs and the nearest branching point (cf. Fig. 4b).

Hierarchical clustering was performed with 1D *k*-means clustering with *k*=2 for all splits, using `KMeans` from the Python package *scikit-learn* (Table 2). First, cells were split into two clusters based on arbour density difference (cf. Fig. 2b). Next, the group with lower arbour density difference was separated by soma size, while the group with the higher arbour density difference was further split based on their asymmetry index. Here, we refrained from further splitting, because the cells in each group displayed highly consistent light responses. Thus, these four groups were used for further analysis.

**Data analysis.** All data were analysed using custom scripts: For data pre-processing, we used IGOR Pro; further analysis and modelling was done using Python and R. All data, scripts and models are available (see links in Table 2).

ROIs: For the dendritic Ca$^{2+}$ responses evoked by light stimulation, we used dense noise recordings to extract ROIs. First, for each recorded field, the standard deviation (s.d.) of the fluorescence intensity for each pixel over time was calculated, generating an s.d. image of the time-lapsed image stack. Pixels brighter than the mean of the s.d. image plus 1 s.d. were considered dendritic pixels. Then, in each recorded field, the time traces of the 100 most responsive dendritic pixels (=100 brightest dendritic pixels in the s.d. image) were extracted and cross-correlated. The mean of the resulting cross-correlation coefficients ($\rho$) served as correlation threshold ($\rho_{\text{Threshold}}$) for each field. Next, we grouped neighbouring pixels (within a distance of 3 μm) with $\rho > \rho_{\text{Threshold}}$ into one ROI. For the dendritic Ca$^{2+}$ signals evoked by current injections, we manually drew 5–10-μm ROIs on the dendrite.

Finally, each ROI's Ca$^{2+}$ trace was extracted using the image analysis toolbox SARFIA for IGOR Pro[64]. A time marker embedded in the recorded data served to

align the traces relative to the visual stimulus with 2 ms precision. All stimulus-aligned traces together with the relative ROI positions on the recorded cell's dendritic arbour were exported for further analysis.

Dendritic receptive fields: Dendritic RFs were estimated using Automatic Smoothness Determination (ASD, ref. [27]), a linear-Gaussian encoding model within the empirical Bayes framework. The relationship between stimulus and response was modelled as a linear function plus Gaussian noise:

$$\mathbf{y} = \mathbf{k}^T \mathbf{X} + \varepsilon, \ \varepsilon \sim N(0, \ \delta^2) \tag{1}$$

where $\mathbf{X}$ is the binary dense noise stimulus (20 × 15 matrix of 30 μm per pixel), $\mathbf{y}$ is the gradient of the Ca$^{2+}$ response, $\mathbf{k}$ is the spatio-temporal RF (STRF) with a time lag ranging from −1000 to 0 ms, and $\varepsilon$ is independent and identically distributed (i. i.d.) Gaussian noise with zero mean and $\delta^2$ variance.

The STRF was then calculated in two steps[27]: First, the ASD prior covariance ($C_{ij} = \exp(-\rho - \Delta_{ij}/2\delta)$, where $\Delta_{ij}$ is the squared distance between any two filter coefficients), controlled by the spatial and temporal smoothness ($\delta$) and scale ($\rho$), was optimised using evidence optimisation. Then, the STRF was estimated by maximum a posteriori linear regression between response and stimulus using the optimised prior. The spatial RF maps shown represent the spatial component of the singular value decomposition of the STRF.

To determine the quality of spatial RFs, contours were drawn on up-sampled and normalised RF maps with different thresholds (0.60, 0.65, and 0.70). The quality was then determined from the number of regions with closed contours, their sizes and their degree of irregularity. The irregularity index was defined as

$$\text{Ii} = 1 - \frac{A_{\text{contour}}}{A_{\text{convex hull}}} \tag{2}$$

with $A_{\text{contour}}$ corresponding the area of a region with closed contour, and $A_{\text{convex hull}}$ the 2D morphology's convex hull. Only data with a good RF (a single contour with Ii < 0.1, $A_{\text{contour}} > 1.8 \times 1000$ μm$^2$, at a contour threshold of 0.60; see Supplementary Fig. 1) were used for further analyses. The cut-off on irregularity (Ii) was set to ensure the RF was approximately round, while the cut-off on $A_{\text{contour}}$ was set to a value much smaller than the RF of a RGC, that still ensured that small, noise-generated random "bumps" were excluded (cf. Supplementary Fig. 1b).

RF Offset distance and angle: RF offset distance was calculated as the linear distance between ROI centre and the geometrical centre of its RF contour. RF Offset angle was calculated as the angle between lines from ROI centre to geometrical centre of its RF contour and ROI centre to dendritic arbour centre.

Dendritic RF overlap index: An RF overlap index (Oi) was calculated as follows:

$$\text{Oi} = \frac{A_o}{A_{\min}[A_1, A_2]} \tag{3}$$

where $A_1$ and $A_2$ are the RF areas of the ROI pair, $A_o$ is the overlap area between $A_1$ and $A_2$, and $A_{\min}[A_1, A_2]$ corresponds to the smaller area ($A_1$ or $A_2$).

Full-field chirp and local chirp: Ca$^{2+}$ traces for full-field and local chirp stimuli were linearly up-sampled (interpolated) to 500 Hz, baseline-subtracted (using the mean of 2500 samples before light stimulus onset) and normalised by the s.d. of this baseline. To estimate the signal-to-noise ratio, we calculated the response quality index (Qi) for both full-field and local chirps as described in Franke et al.[37]:

$$\text{Qi} = \frac{\text{Var}[\langle C \rangle_r]_t}{\text{Var}[\langle C \rangle_t]_r} \tag{4}$$

where $C$ is the T-by-R response matrix (time samples by stimulus repetitions) and $\langle \rangle_x$ and $\text{Var}[]_x$ denote the mean and variance across the indicated dimension, respectively. If all trials were identical, such that the mean response is a perfect representative of the response, Qi = 1. If all trials were completely random with fixed variance, such that the mean response is not informative about the individual trials, $\text{Qi} \propto 1/R$.

Signal correlation: To quantify temporal signal correlation, we cross-correlated the mean Ca$^{2+}$ responses. We noticed that some ROIs with good spatial RFs (see above) displayed low signal-to-noise chirp responses. Hence, we repeated the analysis for Qi > 0.4 or 0.5 with comparable result (Supplementary Fig. 10).

Further temporal analysis: Using the responses to the step part of the chirp stimulus, we calculated a transience index (Ti, for local chirp) and a polarity index (POi, for both local and full-field chirps). Here, only ROIs with Qi > 0.4 were used for the analysis. Before the computation of these indices, the mean traces were binomially smoothed (with 3000 repetitions). Then, 2 s.d. of the baseline (2500 samples of the smoothed trace before light stimulus onset) were used to determine the time of response onset ($T_{\text{R\_onset}}$) and offset ($T_{\text{R\_offset}}$).

Ti was calculated as

$$\text{Ti} = 1 - \frac{T_{\text{R\_offset}} - T_{\text{R\_onset}}}{T_{\text{stimulus}}} \tag{5}$$

where $T_{\text{stimulus}}$ is the stimulus duration.

For POi, data points before and after the response (see above) were set to zero, before calculating:

$$\text{POi} = \frac{\sum_{t=0}^{b} r(t + t_{\text{stim on}}) - \sum_{t=0}^{b} r(t + t_{\text{stim off}})}{\sum_{t=0}^{b} r(t + t_{\text{stim on}}) + \sum_{t=0}^{b} r(t + t_{\text{stim off}})} \quad (6)$$

Where $b = 3$ s, $t_{\text{stim on}}$ and $t_{\text{stim off}}$ are the time points of light stimulus onset and offset, $r(t)$ is the mean response at time $t$. For ROIs responding only to the light-onset, POi $= 1$, whereas for ROIs only responding during the light-offset, POi $= -1$.

Backpropagation: For each ROI, Ca$^{2+}$ traces were normalised by the s.d. of the baseline (first 5 s of the raw trace). To compensate for changes in baseline, the mean of the 5-s trace before the onset of the current step was then subtracted from the current-evoked Ca$^{2+}$ response. The dendritic Ca$^{2+}$ response (area under the curve) and somatic action potential count was determined within a time window of ~1.6 s, starting from the onset of the 500-ms current step.

**Statistical analysis**. We used Generalised Additive Models (GAMs) to analyse the relationships of RF size vs. dendritic distance; RF offset vs. dendritic distance; RF overlap vs. dendritic distance and dendritic angle; temporal correlation vs. dendritic distance and dendritic angle (for details, see Supplementary Statistical Analysis). GAMs extend the generalised linear model by allowing the linear predictors to depend on arbitrary smooth functions of the underlying variables[65]:

$$g(\mu) = \beta_0 + f_1(x_1) + \ldots + f_n(x_n) \quad (7)$$

Here, $x_i$ are the predictor variables, $g$ is a link function, and the $f_i$ are smooth functions of the predictor variables. These smooth functions can also depend on more than one predictor variable.

To implement GAMs and perform statistical testing, we employed the *mgcv* package for R (Table 2). Here, for smooth terms we used penalised regression splines. We modelled the dependence of our variable of interest as a single smooth term per cell type for univariate dependencies and a tensor product smooth for bivariate dependencies. The dimension of the basis was set high enough such that the estimated degrees of freedom stayed sufficiently below the possible maximum. Smoothing parameters were selected via the default methods of the package. All models also included a random effect term for cell identity. Typically, we used models from the Gaussian family; for the dependence of RF overlap on dendritic distance and dendritic angle, we instead used a scaled *t*-distribution, as this improved BIC (Bayesian information criterion) and diagnostic plots (see also Supplementary Statistical Analysis).

Statistical significance for differences in the obtained smooths between cell types were performed using `plot_diff` or `plot_diff2` of the *itsadug* package for R (Table 2). 95% confidence intervals (CI) were calculated using the simultaneous confidence intervals (CI) option, excluding the random effect of cell identity.

To analyse the relationship between Ca$^{2+}$ signals and somatic action potential count evoked by current injections, we used linear regression. Model fitting was performed using R's build-in function (`lm`) and the *emmeans* package for R. Statistical significance for the interaction of dendritic Ca$^{2+}$ signal and somatic action potentials and pairwise comparison of slopes were performed using R's build-in function `anova`, `pairs`, and `lstrends` (*emmeans*) (see also Supplementary Statistical Analysis).

**Biophysical model**. To explore the mechanisms underlying dendritic integration in different RGC types, we built a multi-compartmental 1D model. To get precise measurements of dendrite thickness and segment length for tOff alpha and tOff mini cells, we extracted these information from published morphologies of 4ow and 4i RGCs (cf. Fig. 7b), respectively, reconstructed from EM data (http://museum.eyewire.org). Then we mapped the medium values of these parameters to the respective branch order of the model (cf. Fig. 7c). The model was implemented in the NEURON simulation environment[66]. Here, each dendritic portion (between two branch points) was represented as a section in the simulator, which was further divided into multiple segments (compartments) with a maximal length of 7 μm. The 1D model can be characterised by the cable equation,

$$\frac{d}{4\tilde{r}_a} \frac{\partial^2 V}{\partial x^2} = C_m \frac{\partial V}{\partial t} + I_{\text{ion}} - I_{\text{stim}} \quad (8)$$

where $V$ is the voltage across the cell membrane, $x$ is the distance along the cable, $d$ is the dendritic diameter, $\tilde{r}_a$ is the intracellular resistivity, and $C_m$ is the specific membrane capacitance. $I_{\text{ion}}$ represents the sum of four voltage-gated cation currents (sodium, $I_{\text{Na}}$; calcium, $I_{\text{Ca}}$; delayed rectifier potassium, $I_K$; A-type potassium, $I_{\text{K,A}}$), one calcium-activated potassium current ($I_{\text{K,Ca}}$), and one leak current ($I_{\text{Leak}}$). The current dynamics are described following Fohlmeister and Miller[67] as:

$$\begin{aligned} I_{\text{ion}} &= I_{\text{Na}} + I_{\text{Ca}} + I_K + I_{\text{K,A}} + I_{\text{K,Ca}} + I_{\text{Leak}} \\ &= \bar{g}_{\text{Na}} m^3 h(V - V_{\text{Na}}) + \bar{g}_{\text{Ca}} c^3 (V - V_{\text{Ca}}) \\ &\quad + \left( \bar{g}_K n^4 + \bar{g}_{\text{K,A}} a^3 h_A + \bar{g}_{\text{K,Ca}} \right)(V - V_K) \\ &\quad + \bar{g}_{\text{Leak}} (V - V_{\text{leak}}) \end{aligned} \quad (9)$$

The intracellular stimulation current ($I_{\text{stim}}$ in Eq. 8) was the product of a 5000 ms × 200 μm 1D Gaussian noise stimulus and a BC's spatial RF with a

---

**Table 3 Model parameters.**

| Parameters | Values |
|---|---|
| Temperature | $T = 32\,°C$ |
| Intracellular axial resistivity | $R_a = 110\ \Omega\ \text{cm}$ |
| Specific membrane resistance | $R_m = 15{,}000\ \Omega\ \text{cm}^2$ |
| Specific membrane capacitance | $C_m = 1\ \mu F\ \text{cm}^{-2}$ |
| Potassium reversal potential | $V_K = -75\ \text{mV}$ |
| Sodium reversal potential | $V_{\text{Na}} = 35\ \text{mV}$ |

---

**Table 4 Reference distribution of ion channels in cell compartments.**

| Channel type | Conductance in soma [S cm$^{-2}$] | Conductance in dendrites [S cm$^{-2}$] |
|---|---|---|
| $\bar{g}_{\text{Na}}$ | 0.08 | 0.025 |
| $\bar{g}_{\text{Ca}}$ | 0.0015 | 0.002 |
| $\bar{g}_K$ | 0.018 | 0.012 |
| $\bar{g}_{\text{K,A}}$ | 0.054 | 0.036 |
| $\bar{g}_{\text{K,Ca}}$ | 0.000065 | 0.000001 |

---

Gaussian shape (with the width set by $\sigma = 6$ and the centre depends on the current injection location). The stimulation current was injected either at the proximal (at 25 μm from soma) or distal dendrite (at 85% of the total dendrite length from soma), then filtered by a soft rectification function. The magnitude of $I_{\text{stim}}$ was scaled between 0 and 15 nA with a mean of 6.17 nA and an s.d. of 1.88 nA to ensure that the input stimulus would elicit spikes in the soma for all parameter combinations.

Model parameters (Table 3) and channel conductances (Table 4) were taken from Fohlmeister and Miller (ref. [67]). To identify parameters that explain the experimental data, we grid-searched combinations of ion channel densities by multiplying the reference parameters with different scaling factors (for $\bar{g}_K$ and $\bar{g}_{\text{Ca}}$, the scaling factors ranged from 0 to 2, and were incremented by 0.25 each step; for $\bar{g}_{\text{Na}}$, we used [0.1, 1, 2], for $\bar{g}_{\text{K,Ca}}$, [10, 100, 1000, 5000, 9000]).

Voltage changes in the dendrites were read-out at the centre of each dendritic section and used to estimate the local dendritic RF with a maximum likelihood method, then smoothed by a Savitzky–Golay filter (window length of 31; 3rd degree polynomial). The peak amplitudes of dendritic RFs were measured and normalised to the RF at the position closest to the current input.

**Reporting summary**. Further information on research design is available in the Nature Research Reporting Summary linked to this article.

## Data availability
All relevant data are available at https://doi.org/10.5281/zenodo.3708064.

## Code availability
All custom scripts in Python, R and Igor for data processing, statistical analysis, and plotting are available at https://github.com/berenslab/rgc_dendrites. Code for receptive field estimation is available at https://github.com/berenslab/RFEst.

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

## Acknowledgements

We thank Huayu Ding for helping us with single-cell microinjections, Luke Rogerson for help with statistics and discussion, Zhijian Zhao and Gordon Eske for excellent technical support. This research was supported by Deutsche Forschungsgemeinschaft (DFG, EXC307 to T.E. and EXC 2064, Project Number 390727645 to P.B.; BE5601/4-1, BE5601/6-1 to P.B.; EU 42/10-1 to T.E.); NINDS of the National Institutes of Health (U01NS090562 to T.E.); BMBF (01GQ1601 and 01IS18052C to P.B.; 01GQ1002 to K.F.); BWSF (AZ 1.16101.09 to T.B.); MPG (M.FE.A.KYBE0004 to K.F.).

## Author contributions

The study was conceived and designed by T.B, P.B., T.E., and K.F.; Y.R. carried out the two-photon imaging with input from K.F. and T.E.; Y.R. performed the electrical recordings with input from T.S. and T.E.; Y.R. did all data pre-processing as well as data analysis with respect to kinetic aspects of the cells' responses with input from K.F., P.B., and T.E.; Z.H. performed the cell clustering, RF estimation, and modelling with inputs from P.B. and T.E.; P.B. and Z.H. performed the statistical analysis with input from H.B.; Y.R. wrote the first draft of the manuscript; Y.R., Z.H., T.B., P.B., K.F., and T.E. edited the manuscript.

## Competing interests

The authors declare no competing interests.
