## [Peer Review File · Nature Communications]

Reviewers' Comments:

Reviewer #1:

Remarks to the Author:

In this paper the authors image the dendritic activity of different types of ganglion cells in the retina, and measure the receptive field in different dendritic compartment, ie they correlate stimulus and the activity in specific dendritic compartments, which is what I termed below the dendrite receptive field (RF).

They find that, while some types (tOFF alpha) have dendrites that have a localized receptive field that differs from the one measured in the soma, this is not the case for other types, where the receptive field tends to be the same everywhere.

This is a sound and very interesting study suggesting difference in dendritic integration across cell types, and a very impressive piece of work. My comments are mostly about discussing more some questions/ limitations in the study.

Main comments:

1) an important hypothesis is that the authors strongly suggest that the difference between the different types come from presence or absence of backpropagating action potentials. Isn't it something you could see in calcium data ? At least in the cortex, several studies have reported backpropagating APs with calcium imaging. If they cannot be seen with the current technique it would be useful to discuss why, and how this could be achieved.

2) There seems to be a slight discrepancy between the way results are reported in the main text, and the way analysis is performed in Fig 3 and 4. The text suggests there are 'local' RFs in alpha cells and not in other types, but effects are found by pairwise comparison between types. I guess the authors tried to correlate RF size with dendritic distance to soma for each type (for fig 3), or RF overlap with dendritic distance for each type (for fig 4), without referring to comparison across types ? Reading the text, the expectation would be that in both cases the correlation would be significant for alpha and not for others, but presumably this is not what they found, otherwise they would have reported it. I guess this was not significant and the only way to see something was pairwise comparison between types. If this is the case I think this should be mentioned, and the text should be slightly toned down to suggest that dendritic RFs are 'more local' for t alpha than for other types.

3) Given the rich diversity of inputs to ganglion cells that can be obtained by the interaction between bipolar and amacrine cells, as demonstrated by the same authors (Franke et al), it seems unclear that the different types receive the same kind of inputs. In particular, spatial correlation of the inputs might not be similar across types, and this could at least partially explain the different RF sizes and overlap. The authors do mention this hypothesis in the discussion, but saying this is unlikely. There could be different types of inputs that stratify close and have different extents of spatial correlation. Is there any data or paper to substantiate the hypothesis that the inputs to the different types have to be very similar ? Otherwise I would suggest to tone down a bit the claim, especially in the abstract. Dendritic modeling is one explanation, but the fact that it is more parsimonious could be argued unless you have compelling arguments to explain the choice of the different parameters for channel density (which I don't see here).

Minor comments:

Method to estimate RF: what is the impact of the SNR, which is probably low in these recordings ? Is linear RF a good model ? Does this affect the results if not ?

In particular, since one of the results is about RF size, one could wonder if the SNR varies systematically with cell type / dendrite diameter. If so, there could be an iceberg effect that would

make dendrites with better SNR have a bigger RF. Can the authors exclude this ?

Fig 3E shows nice examples of arrows pointing towards the soma but quantification is missing. One possibility would be to correlate the angle of the arrow with the angle between the ROI position and the soma position ? One could expect that there would be much more (anti-)correlation between these two angles for mini than for t alpha. Is this the case ?

Looking at figure 4C, dendritic overlap seems mostly explainable by dendritic distance. Is dependence angular distance significant ? Is there a way to simplify this plot by keeping only one relevant variable ?

Fig 5: if I understand it well the results here are not completely in line with spatial results - some types might be similar from the temporal point of view and different from the spatial point of view. As noted by the authors, there are similar results for tOff alpha and sOff even if tOff alpha has non overlapping RFs and sOff has overlapping ones. Can the authors elaborate on this difference ? It is not clear if this part has an added value or brings more complexity - or it needs clarification.

Still about fig 5, it is a bit counterintuitive that the smaller stimulus (small disc) creates more spatial correlation than the big disc, and one would welcome an explanation on this.

Methods for dendritic receptive fields: what is the rationale for putting a threshold on A_contour ? I guess small RFs correspond to noise ?

Reviewer #2:

Remarks to the Author:

This paper by Ran et al. presents the results of a study in which fluorescent Ca²⁺ indicator imaging was used to assess local integration in the dendrites of alpha-type ganglion cells in the mouse retina. These ganglion cells are well-characterized morphologically and electrophysiologically and therefore are good subjects for this study. The authors' main finding is that Ca²⁺ indicator signals in different dendritic subregions of one type of ganglion cell, the OFF-transient cell, change independently in response to localized stimuli whereas as such signals appear to be correlated over large areas in the other cell types. The authors argue, based on a compartmental model, that such independence of dendritic Ca²⁺ indicator signals can be explained by intrinsic conductances and membrane electrotonic properties. The authors conclude that the electrophysiological properties of different ganglion cell types can endow these types with the ability to encode different features of a visual scene.

My major concern about the study comes from the fact that the authors are using a non-linear assay (fluorescent Ca²⁺ indicator imaging) to assess a non-linear process (dendritic integration mediated in part by voltage-gated channels). How can the authors ensure that the indicator signals are linear throughout the observed range and that differences are not to variability in cell-type (or even inter-dendritic) intrinsic Ca²⁺ buffering? There are a number of control experiments that should be done to address this concern.

Two related points:

-The ball-and-stick model is electrical where as the data constraining it are from indicator imaging. How are the two related? Please clarify.

-What is generating the indicator signals? Back-propagating action potentials? Voltage-gated Ca channels? Ca²⁺-permeable AMPA and NMDA receptors? Some combination of these? Does it matter? Please comment.

Additionally, relevant information from the literature is not addressed by the authors. Notably:

-Zaghloul et al. 2007 studied Y-type (alpha) ganglion cells in the guinea pig retina and noted non-linear integration of stimuli over space in the surrounds of OFF cells. they concluded that a non-linear surround suppressed ganglion cell responses via presynaptic inhibition of excitatory input.

-Murphy and Rieke 2011 demonstrated that the responses of OFF-transient ganglion cells in the mouse retina (the same cells studied here) are shaped by the electrical properties (notably, a transient Ca current) of electrically-coupled amacrine cells. The authors must consider this information given that they find unique responses in the dendrites of these cells. At the very least, the location of electrical synapses in the ganglion cell dendrites could be responsible for the compartmentalization of Ca²⁺ indicator responses observed here. The gap junctions could be shunting electrical signals, preventing them from propagating over large portions of the dendritic tree.

A final point that the authors should consider and address: the receptive fields of cone bipolar cells are larger than the 30 μm pixels—the fundamental stimulus size—used in this study. It would seem that some of the features of the ganglion cell responses could reflect the properties of spatial integration in the bipolar cells.

Minor:

-For clarity, please identify the ganglion cell illustrated in Figure 1 as an OFF-transient cell.

-What do the authors mean by "similar input profiles" (Line 207)? Please clarify.

Reviewer #3:

Remarks to the Author:

Dendritic integration of synaptic inputs is a fundamental process of neuronal signaling. Understanding principles of dendritic computation is critical for delineating the input-output relationship of the neuron. In this manuscript, Ran et al. investigate how visual inputs are processed by dendritic arbors of four types of OFF retinal ganglion cells (RGCs) in the mouse retina. They developed a method to estimate the receptive fields (RFs) of local dendritic segments based on two-photon calcium imaging of OGB-1 during visual stimulation. By sampling multiple dendritic locations of each RGC, they were able to compare the spatiotemporal receptive field properties of dendritic segments with varying distance from the soma within a single RGC, with the knowledge of the entire dendritic arbor morphology of each cell. They found that the four types of RGCs (tOff alpha, sOff, tOff mini, and F-mini Off) exhibit different patterns of local dendritic RF distributions, and differ in the temporal integration pattern of local dendritic signals. To explore the mechanisms underlying this diversity of dendritic computation, the authors use computational modeling to demonstrate that dendritic morphology alone cannot explain the observed differences in dendritic integration. Notably, the inclusion of voltage-gated conductances, together with dendritic morphology, leads to highly specific algorithms of dendritic processing.

The RF mapping of local dendritic segments using calcium imaging and carefully designed data analysis is an elegant and powerful experiment. Together the dendritic arbor reconstruction, the authors generate a comprehensive dataset that can be used to explore various aspects of structure-

function relationships of RGC dendrites. Overall, this is a nice study illustrating the diverse dendritic processing strategies of neuronal cell types, and highlights the mouse retina as an excellent platform for understanding the role of dendritic computation in the context of well-defined functional circuitry. One issue is that the RF properties of dendritic calcium signals in this study are not only shaped by dendritic integration, but also by the patterns of presynaptic inputs. But this is adequately discussed in the discussion. I only have some minor comments.

1. Line 178-180: "Synchronization of dendrites can originate from strong backpropagation of somatic spikes to the 178 dendrites (reviewed in (Stuart and Spruston, 2015)). This is not only expected to increase dendritic RF size but should also shift the RF's centre closer towards the soma."

In this scenario, the dendritic RF size would increase and the RF center would shift towards the center of the dendritic tree instead of the soma, especially when the dendritic arbors are not centered around the soma.

2. In light of the discussion about backpropagation of somatic spikes, it would be helpful to discuss the nature of the OGB signals in this study. Are most of them suprathreshold events such as somatic or dendritic spikes, or are subthreshold depolarizations reliably picked up by imaging?

3. From the dendritic calcium imaging experiments, the authors conclude that in tOff alpha RGCs local RF sizes decrease as a function of distance from the soma while the other for the other RGC types RF size remained constant. It seems rather difficult to identify these trends from the data points in Figure 3c. Perhaps a binned average plot or a regression line could make this more evident. Similar for Figure 3f.

Reviewer #1 (Remarks to the Author):

In this paper the authors image the dendritic activity of different types of ganglion cells in the retina, and measure the receptive field in different dendritic compartment, ie they correlate stimulus and the activity in specific dendritic compartments, which is what I termed below the dendrite receptive field (RF).

They find that, while some types (tOFF alpha) have dendrites that have a localized receptive field that differs from the one measured in the soma, this is not the case for other types, where the receptive field tends to be the same everywhere.

This is a sound and very interesting study suggesting difference in dendritic integration across cell types, and a very impressive piece of work. My comments are mostly about discussing more some questions/ limitations in the study.

We thank the reviewer for appreciating our work and for the helpful comments.

Main comments:

1) an important hypothesis is that the authors strongly suggest that the difference between the different types come from presence or absence of backpropagating action potentials. Isn't it something you could see in calcium data ? At least in the cortex, several studies have reported backpropagating APs with calcium imaging. If they cannot be seen with the current technique it would be useful to discuss why, and how this could be achieved.

To address the reviewer's comment, we performed additional experiments, in which we recorded from individual RGCs in whole-cell patch-clamp configuration. We injected 500-ms current steps (0.1 to 0.5 nA, $\Delta I=0.1$ nA) into the soma while simultaneously recording the somatic voltage electrically and imaging dendritic Ca^{2+} signals (new Fig. 6). We observed that the same current injection systematically evoked larger dendritic Ca^{2+} signals in tOff mini cells than in tOff alpha cells. Moreover, in tOff mini cells, both proximal and distal dendritic Ca^{2+} signals significantly increased with somatic spike count (linear regression, slope=3.03/1.40, $r^2=0.44/0.31$ for proximal/distal dendrites, $p<0.001$, $n=3$ cells), while in tOff alpha cells, we found only a small increase in Ca^{2+} signals with spike count number (slope=0.44/0.24, $r^2=0.08/0.09$, $p=0.034/0.006$ for proximal/distal dendrites, $n=3$ cells).

These data suggest (i) that backpropagation can be detected with our dendritic Ca^{2+} imaging approach and (ii) that backpropagation of action potentials appears to be more efficient in tOff mini compared to tOff alpha cells, hence supporting our hypothesis that the dendritic arbor of tOff alpha cells is more electrically isolated.

We discuss these control experiments in the revised manuscript and show the data in new Fig. 6.

2) There seems to be a slight discrepancy between the way results are reported in the main text, and the way analysis is performed in Fig 3 and 4. The text suggests there are 'local' RFs in alpha cells and not in other types, but effects are found by pairwise comparison between types. I guess the authors tried to correlate RF size with dendritic distance to soma for each type (for fig 3), or RF overlap with dendritic distance for each type (for fig 4), without referring to comparison across types ? Reading the text, the expectation would be that in both cases the correlation would be significant for alpha and not for others, but presumably this is not what they found, otherwise they would have reported it. I guess this was not significant and the only way to see something was pairwise comparison between types. If this is the case I think this should be mentioned, and the text should be slightly toned down to suggest that dendritic RFs are 'more local' for t alpha than for other types.

The reviewer is incorrect in presuming that we did not report correlations because they were not significant. In fact, we analyzed the data in the framework of Generalized Additive Models (GAM; for details, see Methods and Supplementary Statistical Analysis). Here, we fit a GAM for the dependence of receptive field (RF) size as function of dendritic distance with a factor for cell type.

To make the dependence of RF size on dendritic distance more explicit, we now show the smooth terms from the model in revised Fig. 3 – these roughly correspond to a moving average of the data points, statistically controlling for all other factors. These clearly show that the only cell type showing an indication of larger RFs closer to the soma are tOff alpha cells. This model then also forms the basis of the pairwise comparison plots.

3) Given the rich diversity of inputs to ganglion cells that can be obtained by the interaction between bipolar and amacrine cells, as demonstrated by the same authors (Franke et al), it seems unclear that the different types receive the same kind of inputs. In particular, spatial correlation of the inputs might not be similar across types, and this could at least partially explain the different RF sizes and overlap. The authors do mention this hypothesis in the discussion, but saying this is unlikely. There could be different types of inputs that stratify close and have different extents of spatial correlation. Is there any data or paper to substantiate the hypothesis that the inputs to the different types have to be very similar? Otherwise I would suggest to tone down a bit the claim, especially in the abstract. Dendritic modeling is one explanation, but the fact that it is more parsimonious could be argued unless you have compelling arguments to explain the choice of the different parameters for channel density (which I don't see here).

We agree with the reviewer that the synaptic input the studied RGC types receive will certainly not be the same. As detailed in the manuscript, our knowledge on which bipolar cell types contribute to these Off RGCs is sparse and largely restricted to tOff alpha and sOff cells; much less is known about contributions from amacrine cells. On the other hand, these RGCs tap into overlapping IPL strata and therefore can be expected to share some input.

It was certainly not our intention to claim that differences in synaptic input would not play a role. Nevertheless, we think our finding that the observed differences in dendritic integration are consistent with a model using morphology in combination with realistic channel densities highlights the remarkable potential of these properties for tuning dendritic function. Notably, our new backpropagation experiments (new Fig. 6) lend further support to the view that in tOff alpha and tOff mini cells – the RGC pair with the most dendritic overlap in the IPL –, electrical properties play a crucial role in setting up their respective dendritic integration profiles.

We followed the reviewer's recommendation and tuned down the respective statements.

Minor comments:

Method to estimate RF: what is the impact of the SNR, which is probably low in these recordings? Is linear RF a good model? Does this affect the results if not?

To address the reviewer's concern, we simulated the impact of different noise levels on RF size (Rebuttal Fig. R1; for details, see figure legend). The simulation result showed that the variance of RF sizes increased as more noise is added while the mean remains relatively stable. Also, a larger fraction of RFs is filtered out at higher noise levels as they do not meet our quality criteria.

We agree with the reviewer that the real stimulus-response function is likely to be highly nonlinear in nature and, hence, the use of a linear model represents a simplification. Nonetheless, for the purpose of characterizing spatial RFs in this study, we think that a Linear-Gaussian encoding model, which has been widely and successfully used (e.g. Chichilnisky, Network, 2001; deBoer & Kuypers, IEEE T Bio-Med Eng, 1968; Marmarelis & Naka, Science, 1972; Sahani & Linden, Adv Neural Inf

Process Syst, 2003; Park & Pillow, PLOS Comput Bio, 2011; to name only a few), represents a valid approximation. It has been shown theoretically that such a linear model converges to similar results as more complicated linear-nonlinear models, even though the response used is generated by a nonlinear process (Paninski, Network, 2003). Besides, popular nonlinear models, such as the Linear-Nonlinear-Poisson (LNP) model, are designed to capture the dynamics of spike trains and are therefore probably inappropriate for capturing the continuously changing Ca^{2+} dynamics in our study.

In any case, we now provide a comparison between RFs estimated from a Linear-Gaussian model and an LNP model with simultaneous recorded Ca^{2+} , voltage and spiking responses (new Supplementary Fig. S2).

Rebuttal Figure R1 | Simulation of the impact of the signal-to-noise ratio on the measured RF size. The ground-truth 3D RF is modeled as the Kronecker product of the Gaussian bump (as spatial RF) and a time kernel mimicking temporal RFs observed from data. The simulated response is the dot product of the Gaussian white noise and the ground-truth RF, with different levels of noise added. **a**, Spatial RF map (overlaid with RF contours) determined from signals with different levels of noise added (see numbers above maps). **b**, RF size (black curve) and the percentage of “good RFs” (red curve) from $n=200$ measurements as functions of noise level. To identify “good” RFs, we used the same criteria as in the manuscript (see Methods, Dendritic receptive fields).

In particular, since one of the results is about RF size, one could wonder if the SNR varies systematically with cell type / dendrite diameter. If so, there could be an iceberg effect that would make dendrites with better SNR have a bigger RF. Can the authors exclude this ?

As demonstrated in Rebuttal Fig. R1, RFs are not expected to become larger when SNR improves. In addition, we calculated the distribution of the Quality index (Q_i) for chirp responses in tOff alpha, tOff mini and sOff cells (new Supplementary Fig. S9a) and found these Q_i values to cover similar ranges, arguing against substantial SNR differences between these RGC types. Similarly, we did not find any systematic change in Q_i with dendritic distance to soma (new Supplementary Fig. S9b).

Taken together, we do not expect that differences in SNR have strongly biased the quantification of RF size in our study. If requested, we would be happy to also add Rebuttal Fig. R1 to the revised manuscript.

Fig 3E shows nice examples of arrows pointing towards the soma but quantification is missing. One possibility would be to correlate the angle of the arrow with the angle between the ROI position and the soma position? One could expect that there would be much more (anti-)correlation between these two angles for mini than for t alpha. Is this the case?

Thank you for this suggestion. We measured the angle between the lines from the ROI to the dendritic arbor centre and the ROI to its RF center and plotted the number of acute angle (towards the dendritic arbor centre, $<90^\circ$) and obtuse angle (away from the dendritic arbor centre, $>90^\circ$) as a function of dendritic distance to soma (new Supplementary Fig. S5). As expected, in tOff alpha cells, a substantial fraction of ROIs – in particular those on proximal dendrites – showed obtuse RF offset angles. In the other three RGC types, most RF offset angles were acute.

We added this quantification to the revised manuscript (see new Supplementary Fig. S5).

Looking at figure 4C, dendritic overlap seems mostly explainable by dendritic distance. Is dependence angular distance significant? Is there a way to simplify this plot by keeping only one relevant variable?

We realize that the 2D plot is somewhat more difficult to browse, however, since there is a significant interaction between dendritic distance and angular distance, here is no way simplify this plot to a 1D version. Evidence for this comes from fitting Generalized Additive Models (GAM; for details, see Methods) with different degrees of complexity: In a first GAM, we modeled the dependence on dendritic and angular distance with a single smooth term each (akin to showing 1D dependence). In the second, we allowed for an interaction between the two using a tensor-product smooth. R^2 was 0.71 and 0.81 for the first and second model, respectively. The ANOVA F-test for model comparison showed that the second model was significantly better ($p < 0.00001$), confirming that both dendritic and angular distances define dendritic overlap.

Fig 5: if I understand it well the results here are not completely in line with spatial results - some types might be similar from the temporal point of view and different from the spatial point of view. As noted by the authors, there are similar results for tOff alpha and sOff even if tOff alpha has non overlapping RFs and sOff has overlapping ones. Can the authors elaborate on this difference? It is not clear if this part has an added value or brings more complexity - or it needs clarification.

The reviewer is correct: For local chirp stimuli, the temporal correlations of all three RGC types more or less match the expectations from the spatial analysis – i.e. less correlation in tOff alpha, more in tOff mini and sOff cells (Fig. 5d, left). However, for global chirps, sOff cells deviate from this pattern (Fig. 5d, right): Their dendritic correlation drops much more than that of the other two types (Fig. 5e). We consider this a very interesting finding, as it suggests that dendritic integration in sOff cells may strongly depend on stimulus size. One possible reason may be shunting inhibition from amacrine cells that increases for larger stimuli and results in an increasing (relative) isolation of the sOff dendrites (Roska et al., J Neurophysiol, 2006).

In the revised manuscript, we make this point clearer.

Still about fig 5, it is a bit counterintuitive that the smaller stimulus (small disc) creates more spatial correlation than the big disc, and one would welcome an explanation on this.

We do not find this counterintuitive, as smaller stimuli, such as the local chirp, are expected to drive an RGC's circuit only partially. For instance, input from large-field amacrine cells should be much less pronounced for small stimuli. In a recent paper, we showed that in bipolar cells, stimulus size has a strong effect on the temporal structure of their responses due to interactions with amacrine cells (Franke et al., Nature 2017). This already may explain our observations. In addition, note that in these experiments the stimulus was always centered on the dendritic recording site, locally driving the bipolar cells that provide the excitatory input. Results would likely differ when centering the stimulus on the soma.

The revised manuscript now reflects that point.

Methods for dendritic receptive fields: what is the rationale for putting a threshold on A_contour ? I guess small RFs correspond to noise ?

For noisy ROIs, RFs would consist of multiple components even after smoothing, as illustrated in the cut-out from Supplementary Fig. S1 (right). To exclude these ROIs from the analysis, we devised the following procedure: We considered all "bumps" in a RF that were larger than a threshold. This threshold was set to a value much smaller than the RF of a RGC, ~2/3 as large as a RF of an Off BC. If there was only one "bump" in a RF larger than this threshold and this was approximately round, this "bump" was considered the RF of the recorded structure (i.e. a ROI on a dendrite) and its area was considered the RF size. In this case, the smaller "bumps" were considered noise that was simply ignored. If there were multiple "bumps" larger than the threshold the RF was discarded, as then it was not possible to assign one of the "bumps" as the RF. We hope that clarifies our procedure.

We added this clarification to the revised manuscript.

Reviewer #2 (Remarks to the Author):

This paper by Ran et al. presents the results of a study in which fluorescent Ca²⁺ indicator imaging was used to assess local integration in the dendrites of alpha-type ganglion cells in the mouse retina. These ganglion cells are well-characterized morphologically and electrophysiologically and therefore are good subjects for this study. The authors' main finding is that Ca²⁺ indicator signals in different dendritic subregions of one type of ganglion cell, the OFF-transient cell, change independently in response to localized stimuli whereas as such signals appear to be correlated over large areas in the other cell types. The authors argue, based on a compartmental model, that such independence of dendritic Ca²⁺ indicator signals can be explained by intrinsic conductances and membrane electrotonic properties. The authors conclude that the electrophysiological properties of different ganglion cell types can endow these types with the ability to encode different features of a visual scene.

My major concern about the study comes from the fact that the authors are using a non-linear assay (fluorescent Ca²⁺ indicator imaging) to assess a non-linear process (dendritic integration mediated in part by voltage-gated channels). How can the authors ensure that the indicator signals are linear throughout the observed range and that differences are not to variability in cell-type (or even inter-dendritic) intrinsic Ca²⁺ buffering? There are a number of control experiments that should be done to address this concern.

We thank the reviewer for bringing up this question. We agree that recording local voltage using an optical voltage sensor would be preferable. In fact, we tested a couple of voltage sensors, which however, in our hands did not yield usable signals. Although Ca²⁺ indicators have some potential shortcomings, including some non-linearity and the fact that Ca²⁺ from different sources may contribute to the signal, they can provide very useful information.

To evaluate the relationship between voltage and dendritic Ca²⁺ in the studied RGCs, we performed new whole-cell patch-clamp experiments on individual RGCs and recorded somatic voltage and Ca²⁺ in proximal dendrites simultaneously (new Supplementary Fig. S2a). In addition, to approach the recording conditions used in the manuscript, in a few experiments we injected individual RGCs with a Ca²⁺ indicator (as described in the manuscript) and then performed cell-attached recordings of the cell's spiking. In both types of experiments, the RFs determined from the Ca²⁺ responses were almost identical to those determined based on the voltage signal or the spiking pattern (new Supplementary Fig. S2b-d). This suggests that at least for more proximal dendrites, Ca²⁺ signals are a suitable proxy for the voltage signals.

In addition, we found that the gradient (the rate of change) of the recorded Ca²⁺ signals at a given time point is linearly related to the spike rate calculated from the electrical signal (new Supplementary Fig. S3; correlation ranging between 0.55 and 0.82) in both tOff alpha and tOff mini cells. These data suggest that for these RGCs, the Ca²⁺ signal reports spiking activity in a reasonably linear fashion – at least within the tested limits.

These new control experiments indicate that factors, such as Ca²⁺ buffering or Ca²⁺ contributions from other sources (e.g. intracellular stores), appear not to substantially contribute to the Ca²⁺ signals recorded from the studied RGC types – at least when considering more proximal dendrites. Nevertheless, we now point out these potential caveats of using Ca²⁺ as a proxy for changes in membrane potential more clearly in the revised manuscript.

Two related points:

-The ball-and-stick model is electrical where as the data constraining it are from indicator imaging. How are the two related? Please clarify.

This is a misunderstanding. The ball-and-stick model is not fit to the Ca²⁺ data. Instead, we systematically changed the combination of ion channel densities, combining with type-specific morphological parameters, to see what combination of parameters qualitatively reproduced the experimentally observed signal propagation between soma/proximal dendrites and distal dendrites.

The voltage dynamics in the ball-and-stick model is described by a Hodgkin-Huxley like equation, which is purely voltage-gated, as proposed in Fohlmeister et al. (Brain Res, 1990). The model parameters (e.g. ion channel densities) in the original Fohlmeister model were fitted using whole-cell recordings from salamander RGCs. This model has already been successfully applied to describe spiking patterns in mouse On and Off RGCs (Guo, et al, J. Neural Eng., 2016). However, the contribution of specific ion channel densities for the specific mouse RGC types investigated in our study is still unknown. Given that our new supplementary experiments show that Ca²⁺ and voltage are approximately linear related, one can map the two signals to each other (see above).

-What is generating the indicator signals? Back-propagating action potentials? Voltage-gated Ca channels? Ca²⁺-permeable AMPA and NMDA receptors? Some combination of these? Does it matter? Please comment.

Indeed, RGCs have been shown to express AMPA and/or NMDA receptors (Manookin et al., Neuron, 2010; Zhang et al., J Neurosci, 2009; Chen et al., J Neurosci, 2002; Lukasiewicz et al., J Neurosci, 1993). Therefore, it is likely that synaptic activation of Ca²⁺-permeable glutamate receptors contributed to the measured dendritic Ca²⁺ signals. However, earlier work (e.g. Oesch et al., Neuron 2005; Margolis et al., J Neurosci, 2010) as well as our new patch-clamp recordings (see our reply above) suggest that the Ca²⁺ signals largely reflect Ca²⁺ influx through voltage-gated channels and, hence, membrane depolarization. This change in dendritic membrane potential could

result from synaptic input and/or backpropagating action potentials. As our data suggest, these two factors appear to differ between the studied RGC types.

Our revised manuscript now reflects these considerations.

Additionally, relevant information from the literature is not addressed by the authors. Notably:

-Zaghloul et al. 2007 studied Y-type (alpha) ganglion cells in the guinea pig retina and noted non-linear integration of stimuli over space in the surrounds of OFF cells. they concluded that a non-linear surround suppressed ganglion cell responses via presynaptic inhibition of excitatory input.

-Murphy and Rieke 2011 demonstrated that the responses of OFF-transient ganglion cells in the mouse retina (the same cells studied here) are shaped by the electrical properties (notably, a transient Ca current) of electrically-coupled amacrine cells. The authors must consider this information given that they find unique responses in the dendrites of these cells. At the very least, the location of electrical synapses in the ganglion cell dendrites could be responsible for the compartmentalization of Ca²⁺ indicator responses observed here. The gap junctions could be shunting electrical signals, preventing them from propagating over large portions of the dendritic tree.

We thank the reviewer for pointing out these studies. In the revised manuscript, we now discuss how their findings relate to our results (Discussion)

A final point that the authors should consider and address: the receptive fields of cone bipolar cells are larger than the 30 μm pixels—the fundamental stimulus size—used in this study. It would seem that some of the features of the ganglion cell responses could reflect the properties of spatial integration in the bipolar cells.

We are not sure what the reviewer is referring to. If the reviewer is asking if the difference in dendritic RF sizes between RGC type may be simply due to input from distinct BC types with different RF sizes, we do not think this is the case, because all RGC dendritic RFs were significantly larger than BC RFs ($2.36 \pm 1.18 [\times 10^3 \mu\text{m}^2]$; cf. revised Fig. 3c).

With respect to potential spatial processing in BCs: Here, we adjusted the pixel size of the binary dense noise stimulus to be slightly smaller than RFs of single BCs. This allowed to stimulate neighboring BCs differentially and therefore to estimate RGC dendritic RFs at single BC resolution. Because some BC types have larger RF centers than 30 μm , some degree of spatial integration at the BC level cannot be excluded.

In the revised manuscript, we now mention the rationale for using a 30 μm pixel size for noise recordings and discuss the potential contribution of spatial processing upstream of the RGC.

Minor:

-For clarity, please identify the ganglion cell illustrated in Figure 1 as an OFF-transient cell.

Changed.

-What do the authors mean by "similar input profiles" (Line 207)? Please clarify.

We now clarify this in the text.

Reviewer #3 (Remarks to the Author):

Dendritic integration of synaptic inputs is a fundamental process of neuronal signaling. Understanding principles of dendritic computation is critical for delineating the input-output relationship of the neuron. In this manuscript, Ran et al. investigate how visual inputs are processed by dendritic arbors of four types of OFF retinal ganglion cells (RGCs) in the mouse retina. They developed a method to estimate the receptive fields (RFs) of local dendritic segments based on two-photon calcium imaging of OGB-1 during visual stimulation. By sampling multiple dendritic locations of each RGC, they were able to compare the spatiotemporal receptive field properties of dendritic segments with varying distance from the soma within a single RGC, with the knowledge of the entire dendritic arbor morphology of each cell. They found that the four types of RGCs (tOff alpha, sOff, tOff mini, and F-mini Off) exhibit different patterns of local dendritic RF distributions, and differ in the temporal integration pattern of local dendritic signals. To explore the mechanisms underlying this diversity of dendritic computation, the authors use computational modeling to demonstrate that dendritic morphology alone cannot explain the observed differences in dendritic integration. Notably, the inclusion of voltage-gated conductances, together with dendritic morphology, leads to highly specific algorithms of dendritic processing.

The RF mapping of local dendritic segments using calcium imaging and carefully designed data analysis is an elegant and powerful experiment. Together the dendritic arbor reconstruction, the authors generate a comprehensive dataset that can be used to explore various aspects of structure-function relationships of RGC dendrites. Overall, this is a nice study illustrating the diverse dendritic processing strategies of neuronal cell types, and highlights the mouse retina as an excellent platform for understanding the role of dendritic computation in the context of well-defined functional circuitry. One issue is that the RF properties of dendritic calcium signals in this study are not only shaped by dendritic integration, but also by the patterns of presynaptic inputs. But this is adequately discussed in the discussion. I only have some minor comments.

We thank the reviewer for appreciating our study.

1. Line 178-180: "Synchronization of dendrites can originate from strong backpropagation of somatic spikes to the 178 dendrites (reviewed in (Stuart and Spruston, 2015)). This is not only expected to increase dendritic RF size but should also shift the RF's centre closer towards the soma."

In this scenario, the dendritic RF size would increase and the RF center would shift towards the center of the dendritic tree instead of the soma, especially when the dendritic arbors are not centered around the soma.

We thank the reviewer for pointing this out. The reviewer is correct in that for an asymmetric cell, such as the F-mini^{Off} cell in this study, the RF measured at the soma is not centered on the soma but on the dendritic arbor centre. If backpropagation was strong in such a cell, the dendritic RFs would indeed shift towards the dendritic arbor centre. This is consistent with what we observed in F-mini^{Off} cells: Their dendritic RFs systematically shift towards the dendritic arbor center, reflected in the u-shaped RF offset curve in Fig. 3f.

We made this statement more precise in the revised manuscript.

2. In light of the discussion about backpropagation of somatic spikes, it would be helpful to discuss the nature of the OGB signals in this study. Are most of them suprathreshold events such as somatic or dendritic spikes, or are subthreshold depolarizations reliably picked up by imaging?

OGB-1 is a synthetic Ca²⁺ indicator with high Ca²⁺ affinity ($K_D = 170$ nM; Invitrogen) and comparatively fast kinetics (Hendel et al., J Neurosci, 2008). Previous studies showed that OGB-1

allows to detect single action potentials as well as bursts (Hendel et al., *J. Neurosci.*, 2008; Nevian and Helmchen, *Pflug Arch Eur J Phy*, 2007; Kerr et al., *PNAS*, 2005) as well as subthreshold events (Nevian and Helmchen, *Pflug Arch Eur J Phy*, 2007).

To experimentally investigate the nature of OGB-1 signals in our experiments, we performed simultaneous recordings of somatic voltage and Ca^{2+} (new Supplementary Fig. S3). This data revealed that the amplitude of somatic OGB-1 signals scale linearly with the number of somatic spikes, indicating that OGB-1 signals correspond to a good approximation of across membrane voltage. In addition, these experiments demonstrated that subthreshold events are also evident in proximal dendritic Ca^{2+} changes (Rebuttal Fig. R2). Thus, we are confident that OGB-1 allows us to pick up both supra- and sub-threshold events in RGCs.

The revised manuscript now reflects this point.

Rebuttal Figure R2 | Ca^{2+} signal changes reflect sub-threshold depolarization. Simultaneously recorded somatic voltage and Ca^{2+} traces (proximal dendrite) during the presentation of the dense noise stimulus. Red shading highlights Ca^{2+} events associated with subthreshold voltage events (red stars).

3. From the dendritic calcium imaging experiments, the authors conclude that in tOff alpha RGCs local RF sizes decrease as a function of distance from the soma while the other for the other RGC types RF size remained constant. It seems rather difficult to identify these trends from the data points in Figure 3c. Perhaps a binned average plot or a regression line could make this more evident. Similar for Figure 3f.

We now add the Generalized Additive Models (GAM) fitted curves to Figure 3c,f.

Reviewers' Comments:

Reviewer #1:

Remarks to the Author:

The authors have addressed all my concerns, sometimes beyond what I expected. The new fig 6 is very interesting and makes the paper even stronger. Congratulations for the very nice work.

Reviewer #2:

Remarks to the Author:

The authors have revised the manuscript along the lines suggested. New control experiments and better explanation of nuances and caveats of the research (as well as inclusion of a few important citations) have strengthened the paper. I have no additional comments.

Reviewer #3:

Remarks to the Author:

The authors have adequately addressed my comments.